# Conserved hydrophilic checkpoints tune FocA-mediated formate:H$^+$ symport

Christian Tüting [1,2,8], Kevin Janson [1,2,8], Michelle Kammel [3,8], Christian Ihling [4,5], Jana Lorenz[4,5], Fotis L. Kyrilis [1,6], Farzad Hamdi [1,2], Christopher Erdmann[3], Andrea Sinz [4,5], R. Gary Sawers [3] ✉ & Panagiotis L. Kastritis [1,2,6,7] ✉

FocA belongs to the widespread, evolutionarily ancient formate-nitrite transporter (FNT) family of pentameric anion channels and translocates formic acid bidirectionally. Here, we identify compartmentalized polarity distribution across the complete FocA pore structure – resolved at 2.56 Å – mirrored against a two-fold axis with H209 at its center. A FocA-H209N variant that exhibits an efflux-only channel-like function in vivo reveals a density consistent with formate located directly at N209, abolishing the channel's amphiphilicity. Pyruvate formate-lyase, which generates formate, orients at the cytoplasmic face where formate delivery is regulated by conformational changes in the FocA vestibule. Comparisons with other FNTs suggest a tuning mechanism of formate-specific transport via checkpoints enriched in hydrophilic residues.

The pentameric FNT family is evolutionarily ancient and widespread in bacteria, archaea, and protists and facilitates translocation of monovalent anions through a pore present in each protomer[1]. Despite their ubiquity and reports of 19 structures from 6 micro-organisms to date[2–8], how directional flow of anions might be achieved remains elusive, despite mechanistic proposals[9,10]. FNTs, including the formate channel FocA[2–4], the nitrite transporter NirC[6], the hydrosulfide transporter HSC[5], and the lactate transporter *Pf*FNT[7], share a tertiary structure reminiscent of aquaglyceroporin channels[2–4]. They possess an approximately 20 Å-long pore (spanning 1.35 Å and 1.8 Å in width) that connects the cytoplasm and periplasm through funnel-like vestibules[2–4]. Central to the FNT family's function are conserved histidine and threonine residues (H209 and T91 in *E. coli* FocA), both of which are essential for pH-dependent formate:H$^+$ symport into fermenting *E. coli* cells[9–12]. Two models have been proposed for anion uptake[10]: one suggests

proton delivery by H209[9]; the other posits protonation of formate from bulk water[13]. Both models, however, lack empirical validation of charge distribution across the FNT-specific structural adaptations. In addition, how formate is delivered to FocA by cytosolic pyruvate formate-lyase (PflB)[14,15] is still unknown.

Here, we show the high-resolution structure of native *E. coli* FocA, together with the H209N variant, and describe the dynamic pore features that couple to the protonation state. We integrate simulations, cross-linking and docking with PflB, and sequence conservation to delineate the molecular basis of formate/formic acid translocation in FNTs.

## Results and discussion

### Cryo-EM structure of native FocA and H209N variant

To map all amino acid residues lining the FocA pore, we purified FocA from *E. coli* cell membranes (Supplementary Fig. 1A, B). *E. coli*

[1]Interdisciplinary Research Center HALOmem, Charles Tanford Protein Center, Martin Luther University Halle-Wittenberg, Kurt-Mothes-Straße 3a, Halle (Saale), Germany. [2]Department of Integrative Structural Biochemistry, Institute of Biochemistry and Biotechnology, Martin Luther University Halle-Wittenberg, Weinbergweg 22, Halle (Saale), Germany. [3]Institute of Biology / Microbiology, Martin Luther University Halle-Wittenberg, Kurt-Mothes-Str. 3, Halle (Saale), Germany. [4]Department of Pharmaceutical Chemistry & Bioanalytics, Institute of Pharmacy, Martin Luther University Halle-Wittenberg, Kurt-Mothes-Str. 3, Halle (Saale), Germany. [5]Center for Structural Mass Spectrometry, Martin Luther University Halle-Wittenberg, Kurt-Mothes-Str. 3, Halle (Saale), Germany. [6]Institute of Chemical Biology, National Hellenic Research Foundation, Athens, Greece. [7]Biozentrum, Martin Luther University Halle-Wittenberg, Weinbergweg 22, Halle (Saale), Germany. [8]These authors contributed equally: Christian Tüting, Kevin Janson, Michelle Kammel. ✉e-mail: gary.sawers@mikrobiologie.uni-halle.de; panagiotis.kastritis@bct.uni-halle.de

**Table 1 | Cryo-EM data collection, refinement and validation statistics**

| | Wild-type FocA (EMD-51034) (PDB 9G49) (EMPIAR-12189) | Asymmetric wild-type FocA (EMD-52959) (PDB 9I3K) (EMPIAR-12189) | H209N-FocA (EMD-51035) (PDB 9G4D) (EMPIAR-12188) |
|---|---|---|---|
| **Data collection and processing** | | | |
| Magnification | x240,000 | x240,000 | x240,000 |
| Voltage (kV) | 200 | 200 | 200 |
| Electron exposure (e–/Å$^2$) | 30 | 30 | 60 |
| Defocus range (µm) | −1.0 to −3.0 | −1.0 to −3.0 | −1.0 to −3.0 |
| Pixel size (Å) | 0.5918 | 0.5918 | 0.5918 |
| Symmetry imposed | C5 | C1 | C5 |
| Initial particle images (no.) | 2,260,127 | 2,260,127 | 1,150,907 |
| Final particle images (no.) | 302,987 | 302,742 | 69,787 |
| Map resolution (Å) | 2.56 | 2.87 | 2.97 |
| FSC threshold | (FSC = 0.143) | (FSC = 0.143) | (FSC = 0.143) |
| Map resolution range (IQR, Å) | 2.449 – 2.843 | 2.734 – 3.075 | 2.957 – 3.099 |
| **Refinement** | | | |
| Initial model used (PDB code) | 3KCV | 3KCV | 3KCV |
| Map sharpening B factor (Å$^2$) | 0 (not modified) | 0 (not modified) | 0 (not modified) |
| Model composition | | | |
| Non-hydrogen atoms | 2106 | 10350 | 2093 |
| Protein residues | 276 | 1360 | 275 |
| Ligands | 0 | 0 | 0 |
| B factors (Å$^2$) | | | |
| Protein | 11.31/79.76/28.95 | 23.05/118.84/55.63 | 31.67/101.10/52.42 |
| R.m.s. deviations | | | |
| Bond lengths (Å) | 0.011 | 0.004 | 0.004 |
| Bond angles (°) | 1.556 | 0.969 | 0.963 |
| Validation | | | |
| MolProbity score | 1.64 | 1.56 | 1.51 |
| Clashscore | 8.22 | 7.47 | 9.67 |
| Poor rotamers (%) | 1.08 | 1.56 | 0.90 |
| Ramachandran plot | | | |
| Favored (%) | 98.91 | 99.48 | 99.63 |
| Allowed (%) | 1.09 | 0.52 | 0.37 |
| Disallowed (%) | 0.00 | 0.00 | 0.00 |

overexpressing FocA displays normal growth and formic acid efflux, confirming FocA function[16]. Thereafter, 3665 micrographs were collected (Table 1 and Supplementary Figs. 1C–E and 2), displaying clear side and top views of the pentameric protein complex (Fig. 1A, B and Supplementary Figs. 1D, E and 2), leading to high-quality 2D class averages (Fig. 1B and Supplementary Figs. 1E and 2). The final FocA structure was reconstructed from 302,937 single-particles, achieving a resolution of 2.56 Å (FSC = 0.143, Table 1 and Supplementary Fig. 3A–H), and showing a thorough sampling across orientations (Sampling Compensation Factor (SCF) = 0.87, Supplementary Fig. 3H), with the local resolutions being ~2.3-2.7 Å (Fig. 1C and Supplementary Fig. 3G). The overall structure exhibits differences to the previously published *E. coli* FocA crystal structure[2] as the all-atom root-mean-square deviation (RMSD) is 2.5 ± 0.2 Å. Our structure of the native *E. coli* FocA channel at high resolution (Fig. 1C–E) shows significant rearrangements (Supplementary Fig. 4A–C) and a nearly fully resolved *N*-terminus (Supplementary Fig. 4A–C), which is unique compared to other FNTs[2–8]. In addition, compared to HSC's pore-forming *N*-terminus and *Pf*FNT's β-barrel, FocA's *N*-terminus forms a loop parallel to the pentameric cytoplasmic face, creating the flattest surface among known FocA structures (Supplementary Fig. 5), and generally among

FNTs (Supplementary Fig. 6). This flatness of the surface is imposed by the entirely flat positioning of the *N*-terminal α-helix on the cytosolic face of the FocA pentamer, as compared to other FNTs (Supplementary Figs. 5 and 6).

Resolving the complete FocA structure allows us to trace the formate channel in the native assembly (Figs. 1D, F and 2). The channel is constrained by mainly hydrophobic residues, especially phenylalanine, whereas the periplasmic side displays a partially positive charge, conveyed by lysine (K68 and K165) and histidine (H155) residues (Fig. 1F). These features of the periplasmic surface are in line with its suggested role in formate attraction[11,13]. The crucial H209 faces the pore with its *pros*-nitrogen (N-1) (Fig. 2A), which is structurally conserved among the FNT family, and as such represents the only charged residue within the hydrophobic pore[2–8] (Supplementary Fig. 7). The formate channel around this key residue is highly constrained: while the H209 is coordinated by an interaction with T91 ($d_{vdw}$ = 3.6 Å), which supports in vivo data indicating that this represents the open channel[9,10], the channel's path itself is sterically confined by two opposing phenylalanine residues (F75 and F202), forming a spatial gate (Fig. 2A). Comprehensive channel analysis (see Methods) validated the described amphiphilic properties previously elucidated[2]. It

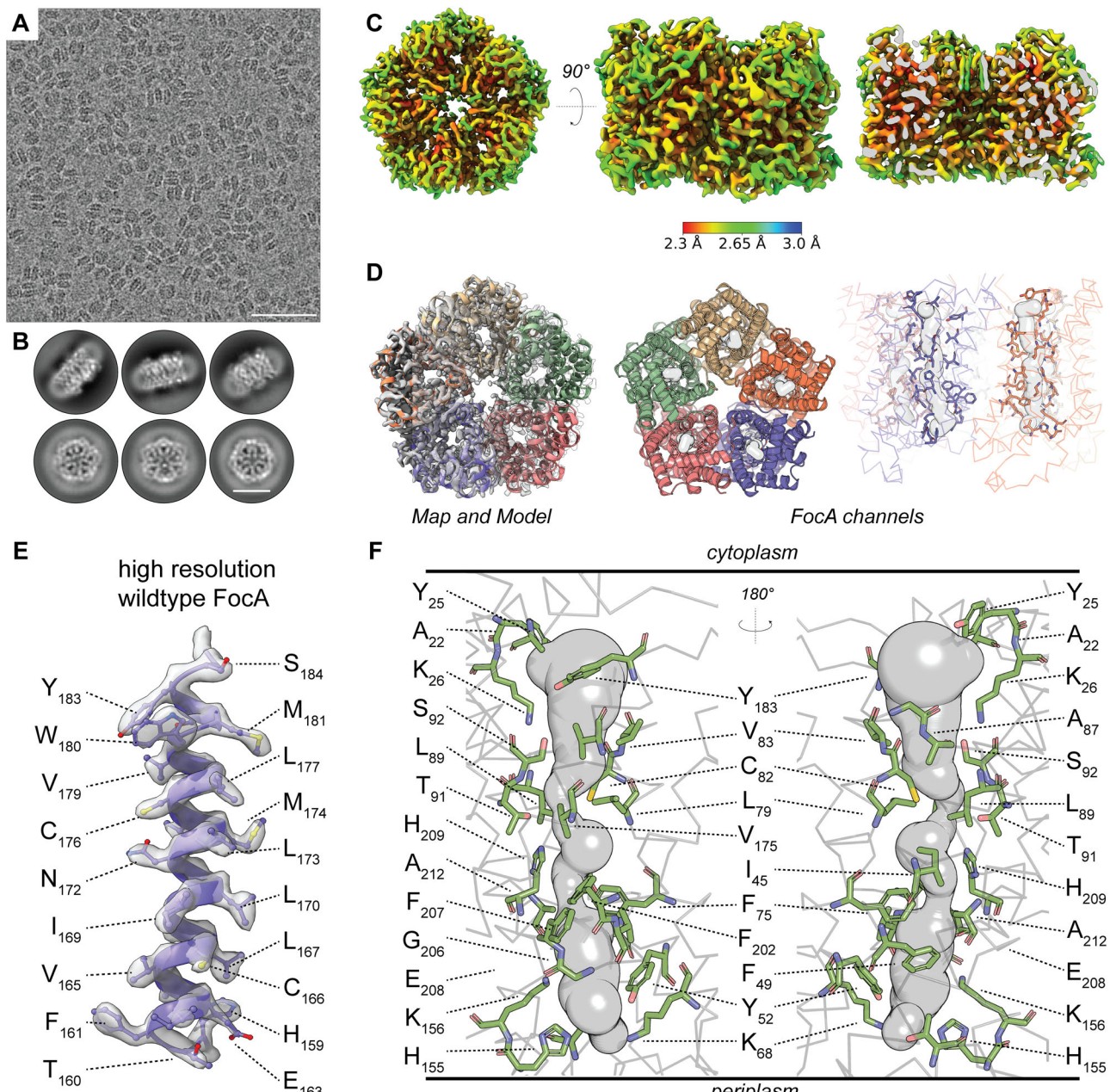

**Fig. 1 | High-resolution cryo-EM structure of full-length FocA. A** Representative micrograph of the wild-type FocA dataset of 3665 micrographs, all of which are available at EMPIAR (EMPIAR-12189 [https://www.ebi.ac.uk/empiar/EMPIAR-12189/]). Pentameric FocA, solubilized by mild detergents, is visible in top and side views. Scale bar is 50 nm. Potential decameric forms arise during membrane solubilization and purification[32]. **B** Exemplary 2D classes of wild-type FocA. Protein structure is clearly discernible, and bound detergent molecules form a delocalized low-resolution density around FocA. Scale bar is 60 Å. **C** Local resolution map of the C5-reconstructed FocA.

Resolution ranges from 2.3 Å within the protein to 2.7 Å at the protein-membrane interface. **D** FocA map and model overlay. The model could be built with high confidence; The formate/formic acid translocation channels indicated as gray spheres are within each protomer. **E** Side chain resolvability, displayed by the α-helical structure. All sidechains could be recapitulated in the derived density. Density threshold is contoured at σ = 0.27. **F** Identification of the formate channel. Sphere diameter corresponds to the channel width. Channel-forming residues are labeled.

further revealed that H209 is crucial to introduce a functionally localized region of polarity (Fig. 2B), which has also been revealed by earlier crystal structures[2–4]. Protonation of H209 could attract formate by altering the local electrostatic potential[4,9,10]. This is supported by physiological evidence, which indicates that pH changes can reverse formate translocation[9,10,16]. In addition, side chain flexibility can modulate the effective pore diameter, as illustrated by comparing the "radius", defined by the side chain atoms, with the "free radius", based on the main chain atoms (Fig. 1G). The difference between these radii suggests that side chain flexibility can act both as a steric barrier that

shifts dynamically and as a mechanism of substrate selectivity, since the substrate must remain in contact with the side chains to pass. In particular, the Phe-gate of F75 and F202 plays a crucial role in this dynamic channel range (Fig. 2A, B). This channel restriction was also previously described to be present in *Clostridium difficile* HSC and might serve as a selectivity filter for the small molecule substrates[5].

We performed the same biochemical and structural analysis for the FocA-H209N variant because this non-ionizable variant performs only unidirectional formic acid efflux in vivo[11] and asparagine (or glutamine) is the only naturally occurring variant at this position across

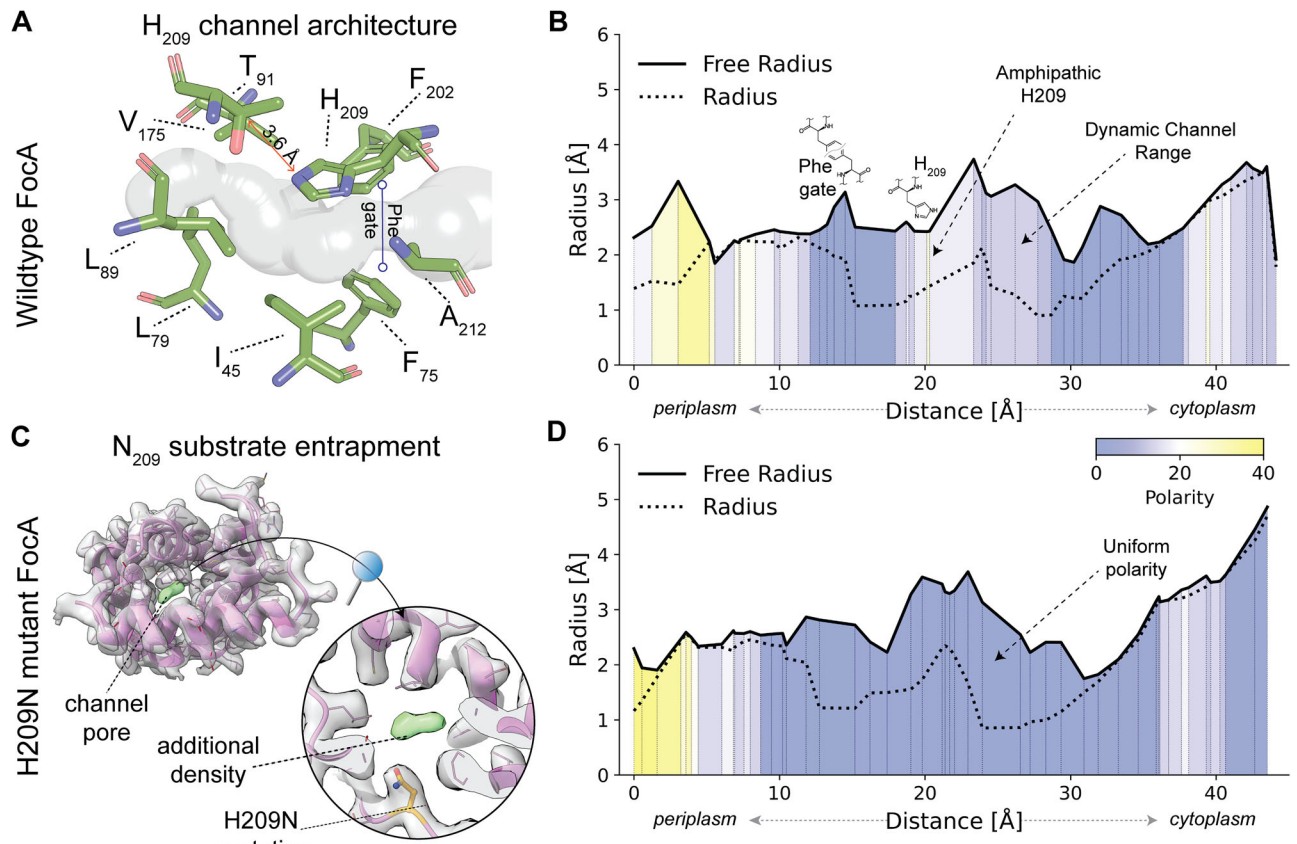

**Fig. 2 | Spatially defined amphipathic channel properties are lost in the H209N variant. A** Zoom-in into the H209 region of the channel. The channel path is curved around this key residue, which is stabilized by a hydrogen bond to T91. Towards the periplasm, the channel width is modulated by two Phe side chains (F75 and F202), forming a spatial gate (Phe-gate). **B** Channel radius and polarity of the native FocA wild-type channel: Radius is defined by all atoms, whereas free radius is defined by only main-chain atoms, approximating flexibility and dynamics of the native structure. The difference between these radii is defined as the dynamic channel range and limited by the Phe-gate towards the periplasm. **C** Detailed view of the channel pore of the FocA-H209N variant. An additional density is identified in proximity to the mutated residue. FocA density is contoured at a 0.2 threshold level, whereas the additional density is contoured at 0.1 for clarity. **D** Channel radius and polarity of the FocA-H209N variant channel, similar to (**B**). Amphipathic properties are absent (yellow regions) in the dynamic channel range, indicating loss of symmetry in the polarity distribution.

thousands of FNTs[10,17]. The resolved structure of FocA-H209N at 2.97 Å (FSC = 0.143, Table 1 and Supplementary Figs. 8 and 9A–H) exhibits minor structural differences compared with the native structure, but the polarity distribution across the pore is disturbed (Fig. 2B, D) due to N209 (Fig. 2D and Supplementary Fig. 10). The isolated FocA-H209N variant revealed the presence of a density close to N209, which would fit various small molecules, including formic acid (Fig. 2C and Supplementary Fig. 10). Capture of a potential translocation intermediate for FocA validates the disturbance of the polarity distribution across the pore (Fig. 2D), while in vivo experiments show that an *E. coli* strain with a mutation in codon 209 of the genomic copy of the *focA* gene, and which synthesizes FocA-H209N, has impaired growth due to reduced intracellular ATP levels (Supplementary Fig. 11), correlating with massive formic acid efflux[10,11]. Past in vitro and in vivo studies using heterologous hosts initially provided valuable insights into the function of FocA[13,18]; however, adopting a homologous system has revealed a more complex regulation of formic acid translocation than previously thought. FocA's in vivo function depends on pyruvate formate-lyase (PflB)[14,15] and the formate hydrogenlyase (FHL) complex[19,20], necessitating a reinterpretation of earlier findings, which were valid at the time but lacked the regulatory context now uncovered (Supplementary Fig. 11). While the H209N FocA variant was found to be non-functional in vitro[13], in vivo experiments in the native environment demonstrate that it operates as an exceptionally efficient formic acid efflux channel (Supplementary Fig. 11), emphasizing the

crucial role of H209 in pH-dependent formate uptake. Immunological analysis of plasmid-encoded FocA-H209N revealed that it is as stable as the native FocA protein[15] and PflB is also stably produced in strain DH4200 (synthesizing FocA-H209N) (Supplementary Fig. 12). These findings demonstrate that the amino acid exchange in FocA-H209N does not affect either synthesis or stability of the protein and has no indirect negative effect on FocA's interaction partner, PflB.

## Molecular dynamics of membrane-embedded FocA channels

To understand the retention of formic acid by residue 209, we performed molecular dynamics simulations, starting with our reported cryo-EM structures (Fig. 3). The set-up included two facing membrane environments, each with one FocA pentamer, with determined chemical composition[21] (Supplementary Fig. 13A), with either formate or formic acid and all H209 protonation states (HSD; Nδ; HSE; Nε; and HSP; Nδ and Nε protonation), as well as the Asn variant (Fig. 3 and Supplementary Figs. 13B and 14A–D). Note that under physiological conditions, formic acid is present in its anionic form, formate. However, we simulated both states for the sake of completeness for efflux analysis from the simulations (Fig. 3 and Supplementary Fig. 14A–D).

By placing the substrate at the additional density identified above (Fig. 2C and Supplementary Fig. 10), the results show preferential retention of formate close to the EM density (Fig. 3A–D) due to the intrinsic property of formate being a hydrogen bond acceptor only (Fig. 3E). Formate exhibits larger displacements (Fig. 3A, B, D), except

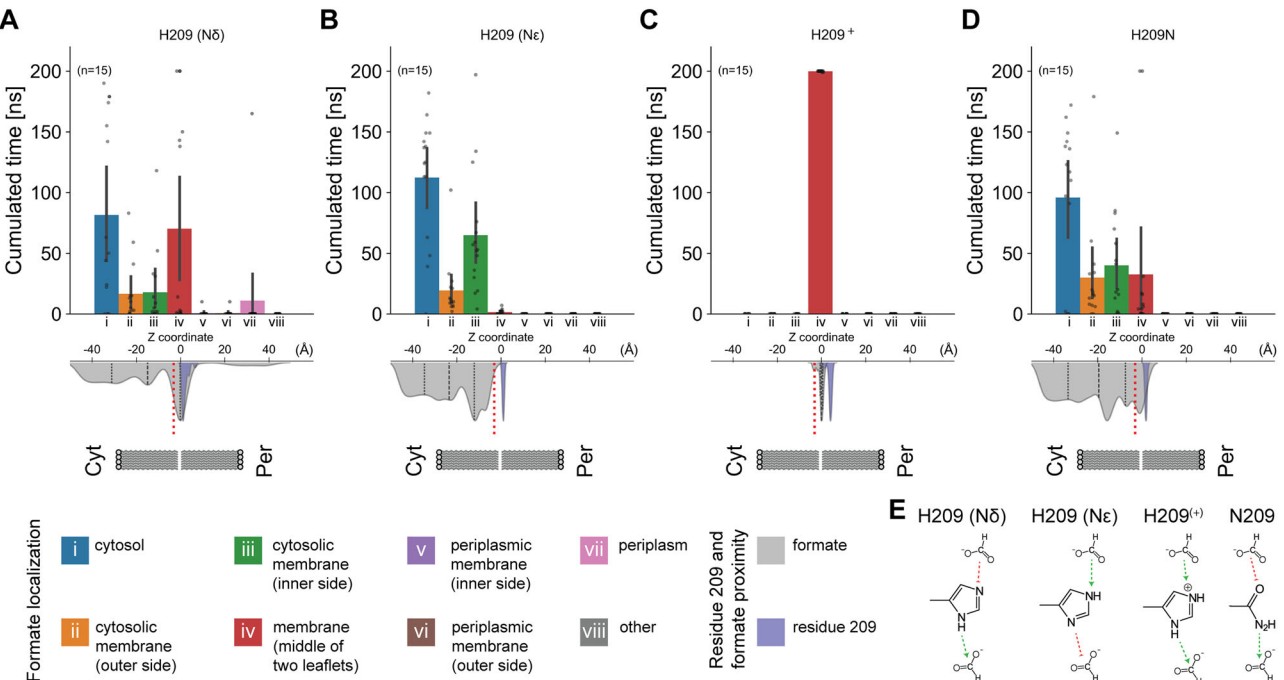

**Fig. 3 | Molecular dynamics simulations reveal protonation state-dependent substrate displacement. A–D** Bar plots of formate localization, and violin plots for substrate (gray) and residue 209 (purple) localization. The membrane is shown as an idealized model, and cytoplasmic (Cyt) and periplasmic sides (Per) are indicated. n refers to individual channel observations (five protomers per FocA pentamer), measured across three independent technical replicates ($n = 15$). Bars represent the mean with error bars showing the standard deviation; individual datapoints are overlaid as scatter points. **A** For the H209 Nδ tautomer, formate is attracted to H209, due to hydrogen bonding, but can also move away, either reaching the periplasm or diffusing into the cytosol. **B** For the H209 Nε tautomer, weaker retention of formate is calculated, and its displacement is monodirectional compared to (**A**) due to tautomeric-specific loss of a hydrogen bond donor (see panel (**E**)). **C** For the H209⁺, anionic formate is strongly attracted and shows no localization variation due to the ionic interaction with cationic H209⁺.

**D** Localization preferences are similar to (**A**), and formate is retained close to the location where the density in the cryo-EM map is resolved. Dotted lines indicate the 25th and 75th quartiles, and the dashed line the data median. Substrate localization for formic acid is shown in Supplementary Fig. 14, per-replicate violin plots are shown in Supplementary Fig. 15. The red dashed line indicates the starting point of the substrate. Statistical details regarding the analysis are described in Methods; "Other" is any position calculated not falling in the 7 other categories. **E** Site-specific hydrogen-bonding propensities. Neutral histidine is in a slow tautomeric equilibrium.

when H209 is charged and is retained with stronger non-covalent bonding because of charge complementarity (Fig. 3C, E). Retention of formate due to the presence of charges recapitulates the known preferential efflux properties of FocA[10,20], which is not affected in the H209N variant (Supplementary Fig. 11). This observation, combined with the preferential retention of formic acid as well (Supplementary Fig. 14A–D), due to its propensities to act as hydrogen bond acceptor and donor (Supplementary Fig. 14E), and the decrease in hydrophilicity in the channel's epicenter (Fig. 2B, D), supports a protonation-dependent translocation of the substrate, explaining our in vivo observations (Supplementary Fig. 11). Another finding is the differential translocation of formate according to histidine's tautomerization state (Fig. 3 and Supplementary Fig. 14 and 15). Proton location in the His imidazole ring may preferentially form a hydrogen bond with formate, inducing its delocalization from His proximity and altering its flux (Fig. 3 and Supplementary Figs. 14 and 15). Cumulative analysis of all protomer channel contacts over all simulations performed showed that interactions around the EM density are highly prevalent. This indicates that this location is important for translocation, promoting the possibility that the captured density corresponds to the substrate (Supplementary Fig. 16). These findings also provide a molecular basis for the previously reported electron densities identified in crystal structures of FocA[2,3].

**Structural asymmetry and flexibility in native FocA**

3D variability analysis of FocA shows flexibility both at the periplasmic and the cytoplasmic sides. At the periplasmic side, flexibility is confined to α-helix A227-L245 (Fig. 4A), which interacts with the α-helix

N142-V157. This α-helix includes residues H155 and K156, which participate in the formation of the periplasmic vestibule (Fig. 1F). The observed flexibility could, in principle, change the dynamic properties of the periplasmic vestibule and therefore, regulate translocation[22].

Variability analysis also shows N-terminal flexibility at the N-terminal helix, as well as the region D98-L110 (Fig. 4A). In addition, these flexible intermediates are coupled to swiveling of the C-terminus (Fig. 4A) and channel "breathing" mediated by two consecutive C-termini (Fig. 4A and Supplementary Movie S1). These substantial changes are localized in the cytosolic and not in the membrane or extracellular part of the channel (Fig. 4A). This aligns with findings of an earlier study[23], in which the short C-terminal helix of FocA was shown to be essential for formic acid translocation by FocA.

Overall, such variability in the cytosolic face is a well-known feature of all FNT channels[2–8]. Absence of the N-terminal helix induces structural changes in its proximity (Supplementary Fig. 17), its partial folding can change the cytoplasmic vestibule architecture (Supplementary Fig. 17A), and it can even block the channel (Supplementary Fig. 17B). Moreover, this is potentially of significance with regard to the pH-dependent flexibility of the N-terminal helix observed in the crystal structure of *Salmonella typhimurium* FocA[4], which exhibits major conformational changes not sampled in our ground-state, physiological pH structure of FocA.

To identify if the FocA pentamer is inherently asymmetric and the observed flexibility at its cytoplasmic side is confined in the protomer, we further analyzed the cryo-EM data for native FocA without imposing any symmetry (Table 1). The model reached a resolution of 2.87 Å

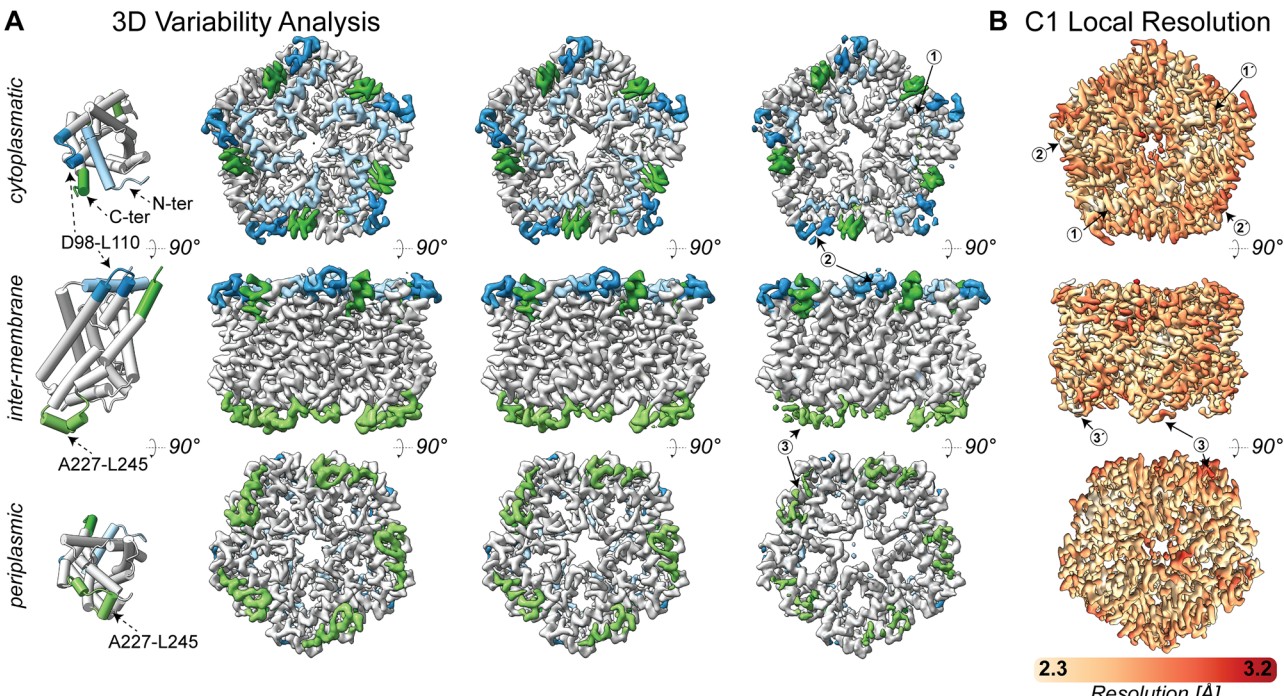

**Fig. 4 | Structural flexibility of the FocA pentamer. A** 3D variability analysis reveals extended flexibility and pore breathing at both the cytoplasmic and peri-plasmic surfaces. At the cytosolic face, the *N*-terminal helix (light blue; (1)) and region D98-L110 (dark blue; (2)) exhibit higher flexibility, as indicated by a dis-continuous density. In addition, the density representing the *C*-terminus (dark green) appears more diffuse, which can be explained by its interaction counterpart, the *N*-terminal helix, undergoing an order-to-disorder transition. At the periplasmic face, region A227-L245 (light green; (3)) displays increased flexibility. The transmembrane region (gray) appears consistent across all sub-volumes, indicating the overall stability of FocA. **B** Local resolution map of the asymmetrically recon-structed full-length FocA. The map is colored from 2.3 Å (light yellow) to 3.2 Å (firebrick). The findings from the 3D variability analysis (**A**) were recapitulated within the asymmetric reconstruction, but regions with elevated flexibility exhib-ited asymmetric behavior. The *N*-terminal helix is clearly outlined at high resolution (1) but appears discontinuous in some regions (1'). Similar asymmetry is observed in region D98-L110 ((2) vs. (2')) and the periplasmic region A227-L245 ((3) vs. (3')).

(FSC = 0.143), with local resolution ranging from 2.7 to 3.1 Å (Fig. 4B). This calculation unambiguously confirms the 3D variability of the symmetric pentamer and shows that individual protomers have dif-ferent degrees of resolvability for all regions identified to be of higher flexibility by the 3D variability analysis (Fig. 4A). These regions were independently identified during the MD simulation to be significantly flexible compared with the overall stable structure of the FocA pen-tamer (Supplementary Fig. 18). Although the complexity of the lipids used in our MD simulations limits the convergence of the membrane to full equilibration, the effects we report concern the equilibrated pro-tein molecule. However, it will be important to investigate in future studies how this asymmetry is influenced by the membrane environ-ment. Such asymmetry can have various implications, which may include: (a) protomers working independently of each other in for-mate/formic acid translocation, as hypothesized previously[4,24]; (b) allosteric communication due to distinct thermal fluctuations of side chains that may reach several kcal mol⁻¹ of entropic contributions[25]; (c) modulating protein-protein interactions via conformational selection mechanisms[26].

**Structural basis for FocA–PflB coupling in formate flux**
Efflux of formic acid is also mediated by a concerted interaction between FocA and the cytosolic enzyme PflB[14,15], with the *N*-terminus playing a potential role[15]. We devised an integrative model of FocA and PflB utilizing cross-linking mass spectrometry (XL-MS) (Fig. 5A and Supplementary Table 1 and Supplementary Fig. 19), the cryo-EM structure of FocA (Table 1), in vivo mutagenesis studies with formate translocation as readout (Supplementary Fig. 20), and the *E. coli* PflB AlphaFold2 structure in complex with its resolved cofactors[27]. To provide further information on the FocA-PflB interface, cross-linking

reactions with the heterobifunctional cross-linker sulfo-SDA[28] and full-length proteins, as well as with an *N*-terminal peptide of FocA and full-length PflB using the homo-bifunctional cross-linker DSBU[29] were performed (Supplementary Table 1). This distance information was integrated with previously published data[14] and utilized as distance restraints in protein-protein docking using the HADDOCK webserver[30]. Combined with our symmetry-agnostic clustering algorithm devel-oped for this study, based on angular sampling (Supplementary Fig. 21A–D; see also Methods for details), PflB confidently orients towards FocA (Fig. 5B). During this procedure, all possible rotations across symmetry points of FocA are sampled, and then the lowest RMSD complex is used as a reference for subsequent clustering. This procedure overcomes the limitations of chain-ID-dependent clustering implemented in HADDOCK.

The conserved flat surface of FocA (Supplementary Fig. 5) acts as an extended docking platform of 3500–4500 Å² for PflB (Fig. 5C). This interaction is mostly polar, and electrostatics play a dominant role (Fig. 5C), in agreement with physico-chemical properties involved in metabolic channeling across transiently interacting proteins[31]. Our model highlights an extension of the FocA channel, where the binding site of coenzyme A (CoA) in PflB is directly above its cytoplasmic vestibule (Fig. 5D). The proposed model underlines a highly transient interaction involving complex interfaces formed by both molecules. For the local formation of the extended channel, only one PflB monomer is involved, together with one of the 5 protomers of FocA, even partially obstructing proximal channels. Such a structure underlines a possible trade-off in delivering substrate to FocA: it is either diffusion-controlled, where several unsuccessful events might occur while all 5 channels are available, or it is channeled directly, where one channel accepts the formate from PflB, while other

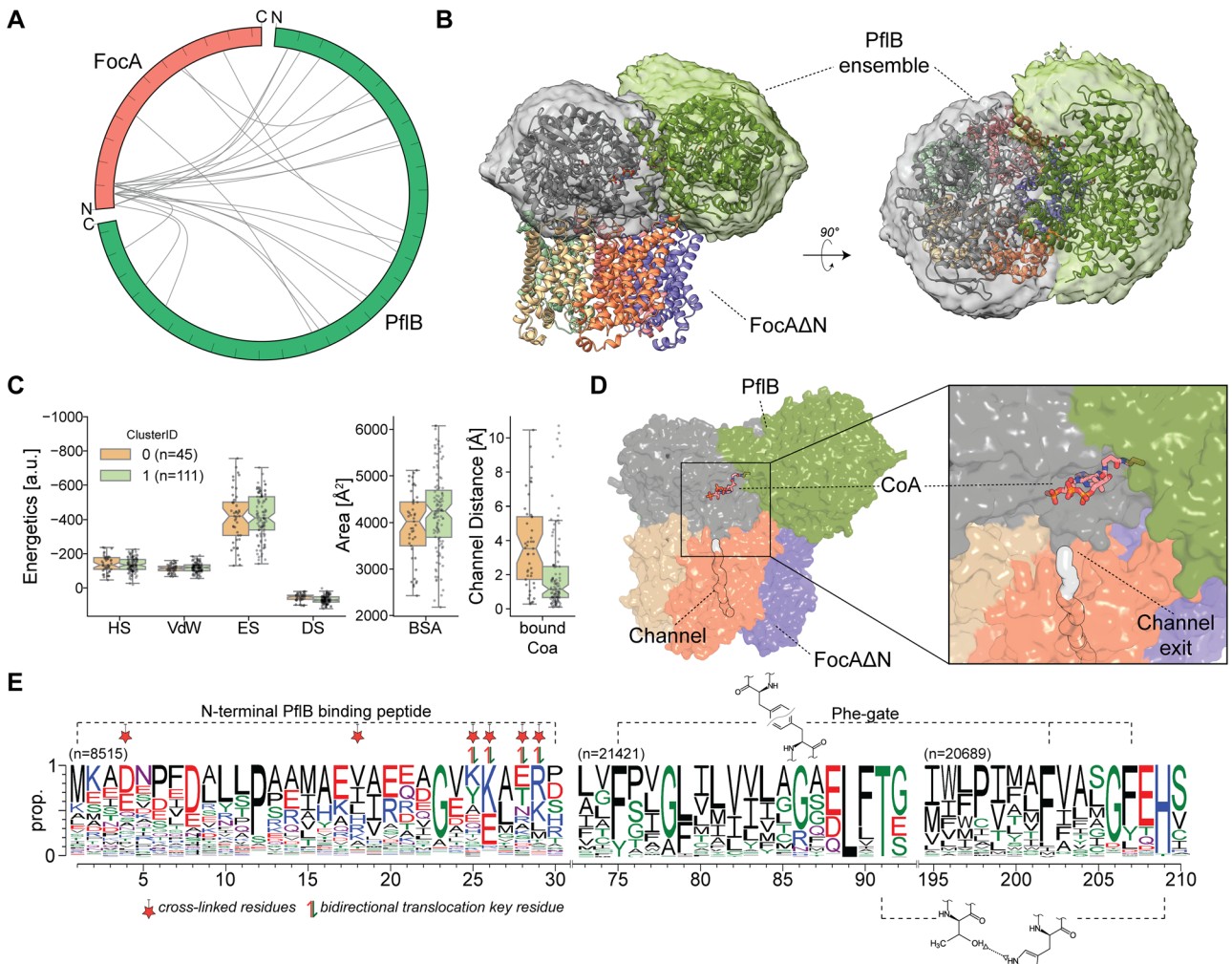

**Fig. 5 | Molecular insights into the FocA substrate translocation in interplay with PflB. A** Circular plot showing extensive cross-linking between FocA's *N*-terminus and multiple regions of PflB. **B** Docking ensemble with all models aligned to FocA, displaying PflB as a confined surface. The top-ranked FocA-PflB complex is shown in cartoon representation. **C** Energetic calculations of the FocA-PflB interface. HADDOCK score (HS), van der Waals (vdW), electrostatics (ES), desolvation (DS) and buried surface area (BSA), as well as the minimum distance of CoA towards the channel axis, are displayed as boxplots, overlaid by the datapoints as scatterplots. The box minima represent the 25th percentile, the box maxima the 75th percentile, the Notch indicates the data's median, whiskers extend to the minimum and maximum value inside a 1.5 interquartile range. n refers to individual water-refined protein models generated in the HADDOCK workflow (200 models in total). Cluster 0 comprises 45 models, and Cluster 1 comprises 111 models. All data points are overlaid as a beeswarm plot; (**D**) View of the FocA-PflB complex reveals direct substrate channeling. CoA is overlaid from the *E. coli* PflB crystal structure (PDB-ID 1H16). The CoA binding site is located at the cytosolic side of the formate channel of FocA; (**E**) Comprehensive sequence analysis of all known formate-nitrate transporters (FNTs) (*n* = 22242), aligned pair-wise to FocA. Sequence conservation is illustrated by WebLogo (https://weblogo.berkeley.edu). *n* indicates the number of aligned sequences with less than 10% gaps. Identified cross-linking sites in FocA are highlighted, as well as key residues of the Phe-gate and the functional amphipathic channel, color-coded with blue (positive), red (negative), and green (hydrophilic).

channels are hindered, or are less efficient. To resolve the in vivo FocA-PflB metabolon unambiguously, future studies of in vivo cross-linking combined with high-resolution in-cell cryo-electron tomography would be relevant.

Comprehensive sequence analysis of all known 22242 FNTs towards *E. coli* FocA revealed clear evidence of sequence conservation at those specific locations (Fig. 5E): The *N*-terminal region (1–30) is aligned in 38% (*n* = 8515) of these FNTs, showing alternate charged (D/E or K/R) and hydrophobic residues, with P12 and G23 being highly conserved. Mutagenesis studies, in which either deletions in the *N*-terminal helix or residue substitutions, were combined with in vivo formate reporter studies, identified residues Y25, K26, T28 and K29 to be crucial for bidirectional formate translocation (Supplementary Fig. 20; the legend details the experimental strategy). These regions are also found to be cross-linked to PflB, highlighting their involvement in the formation of the complex (see Fig. 5A, E).

The channel-forming regions are also highly conserved (93–96% in aligned FNT sequences), in which the key residues T91 and H209, forming the core of the amphipathic channel[2–5], as well as the F75, F202 and F207 residues, forming the Phe-gate, stand out, underpinning the relevance of these residues in modulating these hydrophilic checkpoints via conformational change (Fig. 5E).

Overall, our study has revealed critical, highly conserved checkpoints mediated by polar interactions, which are coupled to localized dynamics. These involve: 1. polarity distribution, also identified previously in FNTs[2–5], but further exemplified here in the full FocA channel; 2. dynamic channel width modulation; 3. conformational dynamics of the *N*-terminal domain to regulate formate/formic acid translocation; 4. and the interaction interface formed by the FocA pore and the PflB active site, identified by previously unreported cross-links between the *N*-terminal region of FocA and its binder, and supported by in vivo mutagenesis experiments relevant for formate/formic acid translocation (Supplementary Fig. 20). All of these molecular locations mediating

translocation of the monovalent anion/cognate acid[12] are highly conserved and can be found in the majority of FNTs[2–8] (Fig. 5E), forming a basis to understand the molecular determinants of H⁺ symport-mediated anion translocation in the widespread superfamily of anaerobic FNTs.

## Methods

### Preparation of membrane fractions and protein purification

Overproduction of native FocA and FocA-H209N, both carrying a C-terminal StrepII-tag, was performed essentially as described[32] using BL21(DE3) transformed with pfocA3 or pfocAH209N, respectively. Cells were cultivated in rich TB-medium, which was supplemented with 100 µg ml$^{-1}$ ampicillin. Protein overproduction was carried out with the following minor modifications. *E. coli* BL21(DE3) cells were used for the overproduction of StrepII-tagged FocA. Cultures (5–15 L) were incubated at 37 °C with vigorous shaking until an OD$_{600nm}$ of approximately 0.4 was attained, at which point anhydrotetracycline (AHT) was added to a final concentration of 0.2 mg ml$^{-1}$ and incubation was continued for a further 16 h at 30 °C. Cells were harvested by centrifugation at 4250 x g for 15 min and were stored at −20 °C until used. Cells were suspended in 5 ml of buffer per g (wet weight of biomass), including 50 mM Tris-HCl, pH 8.0, 170 mM NaCl, 2 mM MgSO$_4$, which also contained Benzonase® Nuclease (Merck, > 90% purity; 100 U) and 4 mM PMSF. Cells were disrupted by two passages through a French-Press cell (1000 psi) at 4 °C. All subsequent steps were also carried out at 4 °C. Non-lysed cells and cell debris were removed by centrifugation at 27,000 × g for 1 h, delivering the crude cell extract. The membrane fraction was prepared from this extract by centrifugation at 135,000 × g for 2 h. Membrane vesicles were suspended in 20 ml of 100 mM Tris-HCl, pH 8, 150 mM NaCl, 1 mM EDTA, 16 mM dodecyl-β-D-maltoside (DDM, > 99% purity, Roth), and the mixture was incubated overnight at 4 °C. After incubation, the centrifugation step was repeated as above (135,000 × g for 1 h) to remove membrane debris and the resulting supernatant was supplemented with 3 nM avidin (GERBU Biotechnik GmbH, Heidelberg, Germany), and incubation was continued for another 30 min. The solubilized FocA released from the membrane fraction was then loaded onto a 5 ml column containing 2 ml of Strep-Tactin Sepharose matrix (IBA Lifesciences, Göttingen). After elution of pure strep-tagged FocA from the column, performed exactly according to the manufacturer's instructions, the protein solutions were dialyzed against 100 mM Tris-HCl buffer, pH 8, including 150 mM NaCl using Spectra/Por® cellulose membrane tubes with a molecular mass cut-off of 3.5 kDa. The dialysis buffer was exchanged twice. After dialysis, the sample volume was reduced at least 10-fold (< 200 µl) using polyethylene glycol (PEG 4000, Merck) to concentrate the protein to approximately 5 mg l$^{-1}$ for cryo-electron microscopy experiments. The yield of FocA was approximately 0.5 mg l$^{-1}$ of culture. Blue-native (BN)-PAGE of purified, strep-tagged FocA variants was performed as described[15]. Western blot analysis of purified strep-tagged FocA, or of membrane fractions bearing FocA, separated in denaturing SDS-PAGE was performed using anti-FocA antiserum (typically diluted 1:1000) as described[15]. His-tagged PflB was purified exactly as described[14].

### Growth studies and formate dependent metabolic analysis

Anaerobic growth studies were performed in M9-glucose minimal medium. Assessment of changes in intracellular formate levels using the *fdhF::lacZ* reporter involved determining β-galactosidase enzyme activity, but performed with the modifications described[15]. Briefly, these modifications included growing the strains anaerobically in 15 ml Hungate tubes in M9-minimal medium containing 0.8% (w/v) glucose and cultures were grown at 37 °C until the exponential growth phase (OD600 nm 0.7–0.9) was attained. Aliquots (100 µl) of cell suspension were then transferred to the wells of a 96-well microtiter plate, and the microtiter plates were stored at − 20 °C until measurements of the kinetics were undertaken. The analysis of extracellular formate level by

HPLC was performed exactly as described[15]. This involved collecting 1.5 ml of cell suspension once the OD600 nm of the culture had reached ~ 0.8. Cells were separated from the culture supernatant by centrifugation through silicon oil (medium-viscosity, PN200, Roth) at 12,000 × g for 5 min. The thus derived supernatant was separated by anion-exchange chromatography on an Aminex HPX-87H column (300 × 7.8 mm, polystyrene- divinylbenzene, 9 µm particle size, and 8% cross-linkage) with the Micro-Guard Cation H+ Refill Cartridge precolumn (polystyrenedivinylbenzene, both columns from Bio-Rad Laboratories) using the high-performance liquid chromatography (HPLC) apparatus (Prominence UFLC; Shimadzu). Formate was eluted with 6 mM H$_2$SO$_4$ at a flow rate of 0.3 ml/min, whereby the column was heated to 42 °C. The amount of formate was quantified by integration of the area of the respective elution peaks, which were monitored at 210 nm and referenced to a calibration curve in the range of 0.1–50 mM formate. Formate concentration was determined in triplicate for each of three biological replicates, and the amount of extracellular formate is presented with the standard deviation of the mean. To analyze formate translocation over time in dependence of growth and to determine PflB levels, strains were grown in serum bottles (100 ml cultures in M9-glucose) and samples were taken as indicated. The respective supernatants were then used to determine organic acid by HPLC and pH (pH meter 765 Calimatic, Knick). Intracellular ATP levels were determined using whole cells and the luciferase-based ATP-Bioluminescence-Assay-Kit CLS II (Roche, Basel Switzerland) following the manufacturer's instructions. The experiments were performed with minimally three biological replicates, and the determined parameters are presented with the standard deviation of the mean.

### Plasmid construction and protein analysis

Plasmid pfocA3, which carries the native *focA* gene with additional codons encoding a C-terminal StrepII-tag on FocA, served as a template for the mutagenesis approaches. Site-directed mutagenesis was performed following the Agilent QuikChange protocol (Agilent Technologies, Waldbronn, Germany). The oligonucleotide primers (IDT BVBA, Interleuvenlaan, Belgium) applied in the experiments are listed in Supplementary Table 2. However, in order to construct the plasmids coding for FocA$_{K26E/K29E}$ a protocol of New England Biolabs using non-overlapping oligonucleotides (Supplementary Table 2), as well as a KLD enzyme mix, was used. All introduced site-specific mutations were verified by DNA sequence analysis of the complete *focA* gene, and the resulting plasmids are listed in Supplementary Table 3. The purity and oligomeric state of the isolated strep-tagged FocA variants were analyzed using polyacrylamide gel electrophoresis. Aliquots (2 µg of protein) of FocA variants were separated in sodium dodecyl sulfate (SDS)-PAGE, including 12.5% or 8% (w/v) polyacrylamide, and subsequently stained with Coomassie Brilliant Blue or using the Pierce™ silver-staining kit (Thermo Fisher Scientific). Alternatively, aliquots (25 µg) of the FocA variants were separated in BN-PAGE. For Western Blot analysis, 25 µg crude extract were investigated using a PflB antiserum, exactly as described[15]. The PflB antiserum used was produced in-house (non-commercially), and its characterization has been reported[33]. The antiserum was used at a dilution of 1:3000.

### Cryo-EM sample preparation and data collection

For the cryo-EM samples, holey carbon support films, type R2/1 on 200 mesh copper grids (Quantifoil, Großlöbichau, Germany) were used. Prior to use, the grids were glow-discharged with 15 mA, grid negative, at 0.4 mbar, and 25 s of glowing time, using a Pelco easiGlow™ apparatus (Ted Pella Inc, Redding, California). A volume of 3.5 µL of a solution with a total protein concentration of 5 mg mL$^{-1}$ was applied onto each grid and subsequently blotted, and flash-frozen by plunging into liquid ethane using a Vitrobot Mark IV System (Thermo Fisher Scientific, Hillsboro, Oregon, USA). The following settings were used: blot force 0, blotting time 6 s, and standard Vitrobot Filter Paper (Grade 595

ash-free filter paper ø55/20 mm). During the whole procedure, a constant temperature of 4 °C and 95% humidity was maintained in the chamber. Subsequently, vitrified grids were clipped and loaded onto a Glacios 200 kV (Thermo Fisher Scientific, Eindhoven, Netherlands) cryo-electron transmission microscope operated in the Kastritis lab, in Halle. Image acquisition was done on a Falcon 3EC direct electron detector in counting mode with a total dose of 51.78 e − /Å$^2$, a defocus range of − 0.8 μm to − 2 μm, and a magnification of 240 kX, resulting in a physical pixel size of 0.5918 Å/pixel. A total of 3665 movies with 30 frames were collected using EPU V 2.9.0.1519REL software (Thermo Fisher Scientific, Hillsboro, Oregon, USA) for wild-type FocA. For H209N-FocA, 2 datasets were acquired of 1073 and 1055 movies, with applied total dose of 63.20 and 59.31 e − /Å$^2$, respectively. Prior to imaging, the electron beam was aligned in parallel and perpendicular to the sample and confined to a diameter of 2.5 μm using a 70 μm condenser aperture. Also, the numerical aperture of the objective lens was restricted to 14.7 mRad using a 100 μm objective aperture.

## Cryo-EM data processing and model building

Both datasets were processed using cryoSPARC (version 4.4)[34]. After patch-based motion correction and contrast transfer function estimation, initial particles were picked by blob picker. For the wild-type FocA, after particle inspection and extraction, a total of 1,953,786 initial particles were used in two-dimensional (2D) classification. The best 2D classes were then selected to re-extract the centered particles. A total of 307,567 particles were extracted and subjected to 2D classification, yielding a particle set of 302,987 particles. After initial reconstruction and homogeneous refinement, a cryo-EM density map with a resolution of 2.58 Å (FSC = 0.143) was reached. The particles along with the motion-corrected movies were used for reference-based motion correction, and final homogeneous reconstruction with C5 symmetry reached 2.56 Å (FSC = 0.143). Based on this EM map, 3D variability analysis[35] with 3 major components was performed, and the first major component showed the N-terminal flexibility coupled with the pore open/closed conformations. Five clusters were then generated, and 3 of those were used to illustrate the conformation changes in Fig. 4A. The same set of particles were used for a 2$^{nd}$ reference-based motion correction and a local refinement with C1 symmetry was performed for the asymmetric reconstruction of native full-length FocA. For the FocA-H209N, 1,150,907 particles were template-picked using the back-projections of the FocA volume, low-pass filtered to 10 Å. After 2D classification, 86,119 particles were selected to be re-extracted for subsequent analysis. Finally, 69,787 particles were re-extracted and non-uniform refinement was applied with C5 symmetry, yielding a reconstruction at 3.16 Å (FSC = 0.143). After C5 symmetry expansion and local refinement with a soft mask generated from this reconstruction, the locally-refined map reached a resolution of 2.97 Å (FSC = 0.143). The AlphaFold model of E. coli FocA (Uniprot ID: P0AC23) was downloaded from AlphaFold DB (https://alphafold.ebi.ac.uk/; accession code: P0AC23) and was rigid-body fitted to the derived maps using UCSF ChimeraX (version 1.7)[36]. The H209N mutation was manually introduced during model refinement. Both FocA structures were rebuilt manually and subjected to real-space refinement using COOT (version 0.9.8.7)[37] and PHENIX (version 1.21rc1)[38]. Due to the C5 symmetry of the complex, only one FocA monomer was modeled in the final structures. These coordinates were further validated using comprehensive validation (Cryo-EM) in PHENIX before deposition in the Protein Data Bank. Figures were created using PyMOL (version 2.6; Schrödinger) and UCSF ChimeraX.

## Molecular dynamics simulations

For molecular dynamics (MD) simulation, the cryo-EM derived symmetric structure was used, and the membrane was generated using the CHARMM-GUI (version 3.8)[39] membrane builder. The lipid composition for E. coli inner membrane during stationary phase was used[21], containing 1-Palmitoyl-2-oleoyl-sn-glycero-3-phosphoethanolamine (POPE): 1-Palmitoyl-2-palmitoleoyl-sn-glycero-3-phosphoethanolamine (PYPE): 1-Oleoyl-2-palmitoleoyl-sn-glycero-3-phosphoethanolamine (OYPE): 1-Plasmenyl-palmitoyl-2-oleoyl-sn-glycero-3-phosphoglycerol (PYPG): 1-Palmitoyl-2-cis-9,10-methylenehexadecanoyl-sn-glycero-3-phosphoethanolamine (PMPE): 1-Pentadecanoyl-2-cis-9,10-methylenehexadecanoyl-sn-glycero-3-phosphoethanolamine (QMPE): 1-Palmitoyl-2-cis-9,10-methylenehexadecanoyl-sn-glycero-3-phosphoglycerol (PMPG) in a 6:17:5:7:32:8:3 ratio (78 total lipids) for the outer leaflet, and 6:17:5:12:20:8:8 (76 total lipids) for the inner leaflet. In addition, pore waters were generated by CHARMM-GUI. All 5 polypeptide chains of FocA were elongated at the N-terminus with MODELLER (version 10.6)[40] to include the full sequence using the AutoModel function without enforcing any secondary structure. For subsequent modifications, the CHARMM36m force-field[41] TIP3 water model were used.

Two membrane systems (membrane and protein structure) were placed facing each other within the simulation box, both in the XY plane. The first membrane system had its Z center at 0, whereas the second membrane system had its Z center at −93 (Supplementary Fig. 13A). By this, a confined cytosolic space was formed, whereas the periplasmic space was confined via the Z-periodicity. The distances between the membranes were big enough (ca. 40 Å) to exclude long-distance interactions. To add 200 mM formate ions into the cytosolic space, the water box was neutralized with 200 mM KCl to atomic coordinates, followed by extraction of the chloride ions between −20 <z< −73, and using PyMOLs align function to align free format (PDB-ID FMT). During system generation using psfgen, the free formate (FMT) was aliased as FORA. Eventually, the simulation box containing both membrane systems and the cytosolic formate was neutralized by 200 mM KCl.

The substrate was positioned near residue 209 (Z ~ − 3.14) for the first membrane and Z = − 90.2. This placement is derived from the cryo-EM map, where the molecule was rigid-body fitted using ChimeraX[36]. Histidine protonation states were calculated using the propKa[42] implementation of PlayMolecules[43], yielding an all HSD tautomer. All input PDB structures were combined using psfgen, with histidines set to HSD except residue 209, which was mutated to HSD, HSE, HSP, or ASN; the substrate was either formate (FORA) or formic acid (FORH). The system was simulated in NAMD 2.14[44] with three consecutive equilibration/minimization steps, followed by a 200 ns production run. In detail, in eq1, the system was first minimized for 10,000 steps using the default conjugate-gradient line-search minimizer of NAMD (function 'minimize'), followed by equilibration with all but solvent and hydrogens constrained. A constraint exponent of 2 ('consexp 2') was used, and restrained atoms were assigned a value of 5.00 kcal*mol$^{-1}$Å$^{-2}$ in the B-factor field. In eq2, the membrane was released; in eq3, the protein side chains were also relaxed. In all equilibration steps, the carbon atom of the bound substrate was constrained to retain its spatial placement but allows for free rotation. Each equilibration was performed for 5 ns, with a 1 fs timestep was used throughout, with non-bonded interactions via a 12 Å cutoff and 10 Å switching distance, and full electrostatics via Particle Mesh Ewald (PME). A Langevin thermostat (damping = 1.0) maintained 303.15 K, with velocities reassigned every 500 steps during equilibration. Periodic boundary conditions were applied with -122 × 122 × 177 Å$^3$ cell dimensions, and group pressure coupling (flexible cell, constant ratio) was used. Bonds to hydrogen were constrained (rigidBonds). After equilibration, a 20 ns production MD was performed under the same conditions, except the time step was set to 2 fs. In addition, the Ca atom of Leu122 was constrained (5.00 kcal*mol$^{-1}$Å$^{-2}$ in the B-factor field) to retain the planarity of the membrane systems. Leu122 was carefully chosen, as it is not involved in the translocation and at the protein-membrane interface. Simulations were run in triplicate, leading to a total of 15 independent channel observations.

## MD data interpretation

MD data was analyzed using the Python package MDAnalysis (version 2.7.0)[45]. First, NAMD trajectory result files (.dcd) together with their respective structure file (.psf) were parsed with the Universe class, enabling programmatic access to the simulation data. For localization analysis, the frame, residue name, residue id, segment id, atom name, and the x, y, z coordinates were extracted into a pandas DataFrame. The time-resolved centers of mass were calculated using the following atoms: formate (FORA): C1, O2, O3; formic acid (FORH): C, O2, O3; residue 209 (HSD/HSE/HSP or ASN): C, N, CA, O. These were obtained via the pandas mean() function. To align the two membrane systems and apply consistent Z-coordinate boundaries, all Z coordinates with $z \geq -46.618$ were rotated relative to the center at $z = -46.618$. For substrate localization, the following logic was applied: if $z \geq 21$, then periplasm; elif $z \geq 15$, then outer periplasmic membrane; elif $z \geq -5$, then inner periplasmic membrane; elif $z \geq -5$, then central membrane; elif $z \geq -15$, then inner cytoplasmic membrane; elif $z \geq -21$, then outer cytosolic membrane; elif $z \geq -21$, then cytoplasm. To identify substrate diffusion through the membrane but outside the protein, every xyz coordinate of every timestep was tested against a cylindrical volume perpendicular to the XY plane, spanning the entire simulation box, with a radius of 40 Å and origin at XY = (0,0). This effectively defined the protein's radial boundary. Timepoints corresponding to transient association with the lipid headgroups ($|z| \leq 21$) were excluded, and replicates in which the substrate fully crossed the membrane (cytosol ↔ periplasm) were removed entirely from the analysis.

For the contact analysis, the MDAnalysis 'distance_array' with a cutoff of 4.0 Å was used. The atomic-level contact information was reduced to a per-residue level. For the time-resolved RMSD calculation, the MDAnalysis 'rms.RMSD' function was used, independently analysis the backbone of both membrane-bound pentamers, as well as both membrane systems. For residue-level backbone RMSD, the coordinates each backbone atom of each frame were extracted from the trajectories, and RMSDs values calculated with custom Python code (see Data Availability for access).

## Chemical cross-linking reaction

Chemical cross-linking between Strep-tagged full-length FocA and His-tagged PflB was conducted with the heterobifunctional photo-/amine-reactive cross-linker sulfo-SDA (Thermo Fisher Scientific). FocA was diluted with 100 mM MOPS, 150 mM NaCl, and 0.03% (w/v) DDM, pH 7.0 to give a final protein concentration of 10 µM. A solution (50 mM) of sulfo-SDA (Thermo Fisher Scientific) in DMSO was added to give a 50-fold molar excess over FocA. FocA was modified with the amine-reactive site of sulfo-SDA by incubation for 1 h at room temperature in the dark. Non-reacted cross-linker was removed using Amicon ultra-filtration units (0.5 ml, 10 kDa cut-off; Millipore) before sulfo-SDA-modified FocA was recovered in 100 mM MOPS, 150 mM NaCl and 1 mM DDM, pH 7.0. A protein preparation of PflB was subjected to buffer exchange using Amicon ultrafiltration units (0.5 ml, 10 kDa cutoff, Millipore). PflB was recovered in 100 mM MOPS, 150 mM NaCl, pH 7.0 and diluted to a concentration of 10 µM. Photo-cross-linking between sulfo-SDA-modified FocA and PflB was induced by irradiation with long-wavelength UV light (maximum at 365 nm, 8000 mJ/cm²).

In addition, chemical cross-linking was conducted between a FocA N-terminal peptide (MKADNPFDLLLPAAMAKVAEEAGVY-KATKHPLKTFGSWSHPQFEK, where residues 1–35 represent the N-terminus of FocA, followed by a short linker (GS) region and the Strep-II tag at the C-terminus; Thermo Fisher) and PflB with the amine-reactive, MS-cleavable cross-linker DSBU[29] (synthesized by Mathias Schaefer, University of Cologne). A protein preparation of PflB was subjected to buffer exchange using Amicon ultrafiltration units (0.5 ml, 10 kDa cutoff, Millipore). PflB was recovered in 100 mM MOPS, 150 mM NaCl, pH 7.0, diluted to a concentration of 10 µM PflB and mixed in equimolar amounts with the synthesized N-terminal FocA peptide (1–45). Afterwards, a 50 mM solution of DSBU in DMSO was added to give a 50-fold or 100-fold molar excess over the FocA peptide. DSBU cross-linking was carried out for 30 min at room temperature. The cross-linking reaction was quenched by Tris-buffer (final concentration of 50 mM). Cross-linking reaction mixtures were enzymatically digested with a mixture of trypsin and GluC (both enzymes from Promega) following an established protocol.

## Mass spectrometry analysis of cross-linked peptides

Peptide mixtures were analyzed by LC/MS/MS on an UltiMate 3000 RSLC Nano-HPLC system (coupled to an Orbitrap Fusion mass spectrometer equipped EASY-Spray™ ion source (all from Thermo Fisher Scientific). Samples were loaded onto a trapping column (Acclaim PepMap C18, 300 µm × 5 mm, 5 µm, 100 Å, ThermoFisher Scientific) and washed for 15 min with 0.1% trifluoroacetic acid (TFA) at a flow rate of 30 µl/min. Trapped peptides were eluted on an self-packed RP C18 separation column (PicoFrit, 75 µM × 250–500 mm, 15 µm tip diameter, packed with ReproSil-Pur C18-AQ, 1.9 µm, 120 Å, Dr. Maisch, Germany) that had been equilibrated with 97% A (0.1% formic acid (FA) in water), 3% B (0.08% FA in acetonitrile). Peptides were separated with linear gradients from 0-35% B over 90 min. The column was kept at 30 °C, and the flow rate was 300 nl/min. Data were acquired in data-dependent MS/MS mode using stepped HCD (high energy collisional dissociation, normalized collision energies (NCE): 27, 30, 33 %) for fragmentation. For data acquisition, each high-resolution full scan (m/z 300 to 1500, $R = 120000$) in the Orbitrap was followed by high-resolution product ion scans ($R = 15000$, minimum charge states 2 + to 6 +) within 5 seconds, starting with the most intense signal in the full scan mass spectrum (isolation window 2 Th); the target value and maximum accumulation time were 50000 and 200 ms. Dynamic exclusion (duration 60 s, window ± 2 ppm) was enabled. Data acquisition was controlled by XCalibur (version 4.0, Thermo Fisher Scientific).

## Identification of cross-linked products

Analysis and identification of cross-linked products were performed with the in-house software MeroX (version 2.0)[46,47]. MeroX was used for automatic comparison of MS and MS/MS data from Mascot generic format (mgf) files. Potential cross-links were manually evaluated. A maximum mass deviation of 3 ppm between theoretical and experimental mass and a signal-to-noise ratio (S/N) ≥ 2 were allowed. Lysine, serine, threonine, and tyrosine residues, as well as protein N-termini, were considered as potential cross-linking sites for DSBU and the first reaction site of sulfo-SDA. As the second reaction site of sulfo-SDA, aspartate and glutamate residues, as well as protein C-termini, were considered. Carbamidomethylation of Cysteine and oxidation of Methionine were taken into account as potential modifications, in addition to three missed cleavage sites for each amino acid Lysine, Arginine, Glutamate, and Aspartate.

## FocA pore analysis and integrative modeling with PflB

For molecular docking of the PflB to the FocA, the HADDOCK 2.2. webserver was used[30]. The cryo-EM structure of FocA was used in both full-length and with an N-terminal truncation, missing the first 30 amino acids (FocAΔN30). The truncation variant was chosen because the N-terminal region of FocA undergoes conformational changes to accommodate PflB (Fig. 2). For PflB (UniProt accession P09373), the dimeric structure was generated using a local installation of AlphaFold2[48,49] (Supplementary Fig. 20A). The multi-chain model of FocA (chains A-E) and PflB (chains A-B) were consolidated into a single chain, adjusting the residue numbering to maintain distinct chain identities by incrementally adding 1000 to the residue numbers. The potential interaction spots, identified by XL-MS, were used as ambiguous distance restraints by including all 10 different interactions

(FocA$_A$-PflB$_A$, FocA$_A$-PflB$_B$, FocA$_B$-PflB$_A$, …). The cross-linked sites and docking distance restraints are listed in Supplementary Table 4. Docking was performed by default parameters using the Guru interface.

## Symmetry-agnostic clustering

HADDOCK clustering relies on the alignment of the water-refined models, based on the chain and residue numbering. Due to the nature of the FocA-PflB models and the ambiguous distance restraint definition, the generated models cannot be aligned using the default algorithms. To overcome this limitation, the following angular sampling approach was developed: From the default HADDOCK scoring function, the restraint violation energy (Eair) was removed, due to high violation of the restraint input. Based on this modified scoring, the highest-ranking model was used as a reference to align all other structures. For this, FocA was aligned and rotated stepwise by 72° to cover all 5 possible symmetry points (Supplementary Fig. 21B). To access model similarity, the RMSD of PflB was calculated, utilizing as reference the PflB of the aligned model in normal and 180° rotated conformations. The rotational alignment with the lowest RMSD values was selected, and the PflB all-vs-all RMSD for the retrieved 200 models was calculated in the same way as before, generating 39800 combinations. For clustering, two models were considered similar if the RMSD of PflB is less than or equal to 7.5 Å, and clustering was done by the Python package NetworkX (Supplementary Fig. 21C, D)[50]. Minimum cluster size was set to 5 % of all water refined models (minimum 10 models should be included).

## Channel analysis and sequence alignments

Analysis of the channel was performed with MoleOnline[51]. The web server was used by default with the following adaptations: Bottleneck Radius: 1.0; Starting point: A 155; End point A 22. For sequence alignment, all annotated Formate/nitrite transporter (InterPro accession: IPR000292) were downloaded from InterPro[52]. In total, 22242 sequences were downloaded, and pair-wise aligned to the FocA sequence using a local installation of Clustal Omega[53]. For plotting sequence conservation, sequences aligned to the FocA reference were filtered to exclude any with more than 10% gaps, ensuring only highly comparable sequences were analyzed. The selected sequences were then visualized using the WebLogo toolkit to graphically represent sequence conservation[54].

## Reporting summary

Further information on research design is available in the Nature Portfolio Reporting Summary linked to this article.

## Data availability

The maps are available in the Electron Microscopy Data Bank: EMD-51034 (wild-type FocA), EMD-52595 (asymmetric wild-type FocA), and EMD-51035 (FocA H209N variant). Atomic models are available in the Protein Data Bank: 9G49 (wild-type FocA), 9I3K (asymmetric wild-type FocA), and 9G4D (FocA H209N variant). Original movies and particle data are available in EMPIAR: EMPIAR-12189 (wild-type FocA) and EMPIAR-12188 (FocA H209N variant). Mass spectrometry data are available in PRIDE: PXD054538 (cross-linking MS data). The input and configuration files from molecular dynamics simulations, as well as the Jupyter Notebooks used for analysis, are available at Zenodo 16939442 [https://doi.org/10.5281/zenodo.16939441]. The following publicly available additional accessions were used within this study: UniProt P0AC23, P09373; AlphaFold DB P0AC23; PDB 1H16 and 3KCV. These resources were used for sequence and structural reference during analysis. Source data is provided as a Source Data file. Source data are provided in this paper.

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

## Acknowledgements

We thank Ioannis Skalidis for assistance in data analysis and our laboratory members for useful discussions. Oliver Trebbin is thanked for help in the purification of FocA. We thank Wiebke Schultze for initial modeling efforts of the FocA protein. We thank Christina Tanja Stöhr for assisting in performing the MD simulations. This work was supported by the European Union through funding of the Horizon Europe ERA Chair "hot4cryo" project number 101086665 (to PLK), the Federal Ministry for Education and Research (BMBF, ZIK program; Grant nos. 03Z22HN23 and 03COV04 to PLK), the Federal Ministry for Economic Affairs and Energy (BMWi, ZIM project KK5096401SK0, to AS), the European Regional Development Funds for Saxony-Anhalt (grant no. EFRE: ZS/2016/04/78115 and ZS/2024/05/187255 to PLK), the International Graduate School AGRIPOLY supported by the European Regional Development Fund (ERDF), the Federal State Saxony-Anhalt (to PLK), funding by DFG (project number 391498659, RTG 2467 to AS and PLK; project number 421152132, CRC 1423 to AS; 514901783, SFB 1664 to AS and PLK; project number SA 494 11/1 to RGS) the region of Saxony-Anhalt, and the Martin-Luther University of Halle-Wittenberg (to AS, PLK and RGS).

## Author contributions

P.L.K., R.G.S., and A.S. conceived and designed the study. M.K. performed the purification, and M.K. and C.E. the functional assays and physiological studies. C.I. performed mass spectrometry. J.L. performed initial XL-MS experiments, A.S. and C.I. performed XL-MS experiments and data analysis. K.J. and F.L.K. performed vitrification. F.H. and K.J. acquired the cryo-EM data. C.T., K.J., and P.L.K. processed data. C.T. and K.J. carried out modeling and map interpretation. C.T. performed the MD simulation and data analysis. C.T. performed the integrative structure calculations. P.L.K. and R.G.S. wrote the manuscript with input from C.T. and all other authors. P.L.K. and R.G.S. supervised the study.

## Funding

## Competing interests

The authors declare no competing interests.
