## [Transparent Peer Review file · Nature Communications]

Conserved hydrophilic checkpoints tune FocA-mediated formate:H⁺ symport

Corresponding Author: Professor Panagiotis Kastiris

Version 0:

Reviewer comments:

Reviewer #1

(Remarks to the Author)

Tüting and co-workers report on cryo-EM structure of the FNT family integral membrane protein FocA from *Escherichia coli* as wild-type protein and as a H209N variant at 2.6 Å and 3 Å resolution, respectively. Using crosslink-MS to support molecular docking, they also present a model for the interaction of the pentameric FocA with the formate producing, cytoplasmic enzyme pyruvate:formate lyase (PflB). A crystal structure of the same protein was reported by Shi and co-workers in 2009 at a resolution of 2.2 Å (3KCU), and a comparison of the experimental maps shows that the EM maps of the present work are equivalent in quality. The structural analysis is carefully conducted and sound and the models are of very high quality. The EM structure aligns with the earlier crystal structure with r.m.s. deviation around 0.3 Å for all atoms, meaning near identity. Furthermore, two orthologs of FocA were reported from *V. cholerae* (3KLY) and *S. typhimurium* (3Q7K).

In their analysis, the authors then assess the role of hydrophilic checkpoints around a central, conserved histidine (H209) that is essential for the unusual bifunctional role of this protein, in that it can act as an efflux channel for neutral formate, or alternatively as a secondary active proton/formate symporter for uptake. The functionalities are connected via a pH-dependent switch mechanism, and in the mentioned H209N variant, only the passive efflux functionality is retained. The authors analyse the polarity of the translocation pore found in each FocA monomer, and describe a central cavity around H209 that is separated from the two exits by a hydrophobic barrier. This feature was already found in the earlier crystal structure, and this should be mentioned more clearly in this manuscript.

An interesting and new aspect is that in the H209N variant, the central cavity seems to have lost its polar properties, which is in line with retaining the function as an export channel for neutral formic acid, but not as an importer for formate anions. Additionally, the mutant structure contains an additional density feature in the channel that can be interpreted as bound formic acid, which at the given resolution cannot be assigned unambiguously.

For the docking model, a series of MS-crosslink restraints were obtained that serve to restrict all HADDOCK solutions correctly to the cytoplasmic face of FocA. This interaction, if real, is most likely highly transient, as not only PflB seems to be a dimer (Fig. 2C), but each monomer of this dimer would obviously only interact with one of the 5 protomers of a FocA pentamer, even partially blocking the others.

On the structural side, a major issue that seems to be missing from this work concerns the structural flexibility of the periplasmic side of FocA proteins. From the first structures on, different conformations for the cytoplasmic funnel were reported, and the *S. typhimurium* structure then even identified different conformations for the N-terminal helix of the protein that would clearly affect conductivity through the individual pores. Earlier work has put quite some emphasis on these features, and it is notable that they are not mentioned here at all.

Instead, the authors now conducted a 3D variability analysis (Fig. 2A) which highlighted flexibility in the C-termini, but not the N-termini. This is not in line with the earlier analysis and might be of interest. It should absolutely be mentioned and discussed in this work.

Related to this, the reported CryoEM structures are refined with C5 symmetry imposed, which averages out any conformational differences between monomers. It would be essential to analyse a careful refinement of this data with C1 symmetry for whether structural asymmetries within one pentamer can be identified. Note that such asymmetry on the cytoplasmic side could possibly be well in line with the symmetry break imposed by the binding of PflB and may even be prerequisite for it. Refining such rather small differences in an otherwise largely symmetric pentamer will likely be

challenging, but seems warranted given the functional implications. Not unexpectedly, a quick run of AlphaFold3 also showed that all solutions, while close to the experiment, are fully symmetrical and do not reflect asymmetric interactions. Interestingly, including PflB in these predictions also consistently places the enzyme close to the cytoplasmic face of FocA, but at a certain distance and not in a consistent orientation.

In summary, this short manuscript reports on a technically sound re-determination of an earlier crystal structure by cryo-EM. It is well written and interesting, and adds valuable information where in the crystal structure protein termini were disordered, and otherwise largely confirms the state of knowledge. Terms such as 'polarity hub' describe findings that were already made in the earlier works and it is not clear to this reviewer that such re-branding is helpful. The issue of symmetry breaks, both within the FocA pentamer and in its predicted interaction with PflB should be elaborated on, and for this the docking experiment could possibly be expanded to the available unsymmetrical structure of *V. cholerae* and *S. typhimurium*.

Reviewer #2

(Remarks to the Author)

In this manuscript, Tüting et al. present a comprehensive study on the structural and functional relationship of FocA from *E. coli*, a member of the formate-nitrite transporter (FNT) family that mediates formic acid transport. The authors utilized cryo-EM to resolve both the wild-type and H209 mutant forms of FocA. Additionally, they employed cross-linking mass spectrometry (XL-MS) to map the interactions between FocA and the cytosolic enzyme PflB, subsequently generating a 3D model based on this data. The structural analysis provided in the manuscript is robust and offers valuable insights into the mechanism of this important FNT family. However, the proposed mechanism could benefit from more direct functional studies to strengthen the conclusions. The revised version of this manuscript would be a strong candidate for *Nature Communications*.

I have the following suggestions and questions for the authors:

Major:

1. The finding that the H209N mutation converts a bidirectional anion transporter into a unidirectional one is intriguing. However, these results seem to contradict transport assays from Eric's lab, which used radiolabeled substrates and demonstrated that the H209N mutation abolishes transport (<https://www.embopress.org/doi/full/10.15252/embj.201695776>). Could the authors provide comments on this?
2. Line 34, "The FocA-H209N efflux-only variant reveals a density consistent with formic acid located directly at N209, breaking local polarity distribution," suggests that formic acid is bound to FocA-H209N. If this substrate is from a native source, it might be worthwhile to use LC-MS or native mass spectrometry to confirm the identity of the substrate bound to FocA-H209N.
3. The functional assays were conducted using whole-cell systems. To provide more direct evidence supporting the proposed mechanism, such as unidirectional transport in the H209N-FocA mutant, the authors might consider using reconstituted proteoliposome-based assays.
4. How does the H209N mutation cause FocA to become a unidirectional efflux transporter? Can authors explain this mechanism using the structural information they have obtained?
5. Including a summary diagram of the proposed mechanism would be highly beneficial for readers.
6. In the pentameric FocA transporter structure solved in this study, structural variability is observed. Is it possible that the protomers may move independently of each other? To better address potential conformational heterogeneity, could the authors clarify if they have attempted to perform C1 symmetry to reduce symmetry restrictions or applied symmetry expansion on the final map (C5) followed by focused classification on a single protomer?
7. Rotamer outliers (table 1) 11.26% and 2.26% in these given resolutions are too high. Authors need to validate their coordinate models.
8. Authors need to include Western blots to indicate the expression levels of FocA and PflB variants, ensuring that observed changes are not due to alterations in protein expression levels across all functional assays.
9. The assembly of the FocA-PflB complex is based on a 1:1 model. Do the authors have any evidence supporting this stoichiometry, such as data from size-exclusion chromatography (SEC)?
10. FocA contains five channels for anion transport. In the FocA-PflB complex model, only one channel appears to be channeling, which could reduce the efficiency of the transport process. Can the authors provide insights on this?

Minor:

1. Line 67. "endogenous *E. coli* FocA channel ..." The term "endogenous" typically refers to protein expression from genomic DNA with a native promoter. If this is the case, the authors are encouraged to provide a more detailed description in the Methods section.
2. Line 315, "FocA pore analysis and derivation of the integrative FocA-pflB complex with". The text should be "FocA-PflB" instead of "FocA-pflB."
3. The raw mass spectrometry data should be deposited to the PRIDE database.
4. Please consider labeling additional side chains around the density in Figure 1H.

Reviewer #3

(Remarks to the Author)

In this manuscript Tüting, Janson and Kammel et al describe the structural characterisation of FocA, a formate nitrate

transporter. This manuscript is overall well written, but is very concise – maybe too concise in places. I would recommend adding some additional details at the point of resubmission. There is potential for re-ordering and for moving figures from the supplementary information into the main text to improve the clarity of the manuscript. My expertise lies in the field of structural proteomics and membrane protein biology, so I do not feel able to comment on the quality of the cryoEM. These experiments are well performed, but the lack of clarity and detail in this manuscript makes it hugely difficult for the reader. I recommend that the authors should revise their manuscript keeping this in mind. I only raise here minor comments, which I'm sure can be addressed simply.

Minor comments:

-line 34, need to define efflux-only variant

-line 35, what do you mean by 'breaking local polarity distribution'

-line 57. The authors refer to Fig S1b where a 10mer is observed by BN-PAGE. It would be useful for the authors to comment here the biological relevance of this higher order assembly, and the precedent (if any) for their formation from previous work.

Figure S4 could be moved to the main text.

-Line 71 – what do the authors mean by 'flattest surface'. The authors should clarify this in the text. Once I see the figure it makes sense, but it would be nicer to see this in the text.

-The authors 'trace the formate channel' in Figure 1E. It is not clear to me however where this fits in the context of the full length protein oligomer. An additional figure, or figure inset may help to orient the reader.

- line 83 – 'Comprehensive channel analysis' what do the authors mean by this?

- Figure 2A – really unclear to see the motions the authors are referring to here.

- line 108 – the authors derive an integrative model for FocA-PfIB. Why did they approach this challenge in this way? Why not an additional cryoEM structural analysis?

Fig S12 – needs more explanation how these data guided the modelling. What do they show? For the non-expert reader this needs much more clarity.

- Figure S13 – the authors state in the text that the structure of PfIB bound to a substrate was used for integrative modelling. Why is there an AlphaFold2 model shown in Figure S13A? How was this used?

- Line 117 – what is their new 'symmetry agnostic clustering algorithm'. Whilst the methods goes into a lot of technical detail about this, a simple few sentences explaining this would be beneficial.

- Why do the authors crosslink between the FocA N-terminal peptide and PfIB? Why were so few inter-protein crosslinks detected for the full length protein? Did the model require all crosslinks, including those from the peptide to reach a satisfactory confidence score?

Version 1:

Reviewer comments:

Reviewer #1

(Remarks to the Author)

In their revision, the authors have extensively and thoroughly addressed the issues raised by this reviewer, and apparently also by the others. This is a very conclusive and informative study of FocA and its interaction with the enzyme PfIB. I fully support publication in Nature Communications and have to further request or criticism. Congratulations to the authors.

Reviewer #2

(Remarks to the Author)

I consider the revised manuscript, 'Conserved Hydrophilic Checkpoints Tune FocA-Mediated Formate:H⁺ Symport,' suitable for publication. The authors have thoroughly addressed the reviewers' concerns with thoughtful responses and necessary revisions. Including MD simulations and reprocessed data enhances the mechanistic insights, significantly strengthening the study, particularly regarding the substrate translocation mechanism.

Reviewer #3

(Remarks to the Author)

The authors have done a thorough job at addressing all my comments and I have no further issues to raise.

Reviewer #4

(Remarks to the Author)

This is a revision of a previously submitted manuscript, and I have been asked to assess the quality of the MD simulations. These have been added to address the EM density, and predict whether it is formate or formic acid.

I have some major concerns about the MD data. These are as follow:

The simulations are very short – 20 ns. And from what I can see no repeats were made for each condition. If this is the case, then there is a chance that the simulations are severely undersampled. I would normally expect at least 200 ns and at least 3

repeats. Ideally 5-10. If there are repeats this should be made clear in the methods, and their densities should be plotted separately to demonstrate the degree of agreement between them.

The analysis presented in 3D is a distribution of all positions of the formate/formic acid from the MD data. To understand the data better, we need to see the time series data too – this will let us know if the simulations have reached any kind of equilibrium or are still converging. Given the very short timescales, the former seems very plausible. I believe that this is a requirement for publishing in Nat Comms anyway.

The formate/formic acid were placed in the EM density. How? How confident of the orientation of the molecule are we? Would this be the same for the different charge states? To address this, I would have personally performed an extensive (100 ns) equilibration where the molecule could rotate but not shift in the Z dimension. But in absence of this, the protocol should be more clearly described as at the moment it is hard to determine what exactly has happened.

The orientation of the formate/formic acid over time should be analysed. As should the contacts it makes with the channel.

What about x/y dynamics? In general the formate/formic acid motions seem confusing to understand. This is something that a video would be useful to show.

In 3D, I'm not too sure about the data:

- Firstly, in all cases the molecule shifts away from the EM density, typically by at least 4 Å. Arguably this suggests that the EM density corresponds to neither formic acid or formate.
- How is the pink residue 209 bar so far from the starting point (red line)? It looks like 6-7 Å but the panels in C look far closer, indeed that the molecule is in contact, so only 3-4 Å away. Does the His contact the molecule?
- the response states: 'Our MD simulations corroborate the presence of formate in the density because its residence times close to residue 209 are higher'. What residence times? I can't see anything like this in the manuscript. His-molecule distances would in part address this, but residence times are very difficult to model in MD, and typically require sampling of multiple binding and unbinding events.
- I've read the results section a few times and I can't tell from the manuscript what the main conclusion is. They state that there's preferential retention of formic acid (agree) which is the opposite of the rebuttal. But then they go on to analyse the formate displacement and not formic acid. So which state of the molecule does the MD support?

In S14 "the MD simulations corroborate the asymmetric nature of the pentamer". Please explain. The periplasmic regions have a different RMSD than the cytoplasmic? Firstly, this can't easily be seen in the current graphs at all. Second, I don't see how the dynamics of the loops in the protein is of relevance to substrate transport at the center of the channel. Perhaps I'm missing something here.

Aside from these concerns, I have some smaller changes that are needed:

- CHARM-GUI > CHARMM-GUI.
- Please spell out the lipid names. For instance, what is PYPG?
- Note that this lipid composition is far too complex for a 20 ns simulation – indeed even a 1 μs simulation would likely not be sufficient to equilibrate the membrane. This makes comparison between simulations more difficult, as the local lipid composition will not be the same. This is why model membranes are typically used (e.g. POPE/POPG).
- The protein chains were elongated using modeller. How were the new sections modelled – as helix, loop, turn etc? The model should be made available for download by the readers.
- Was it CHARMM36 or CHARMM36m? This needs a reference.
- line 385 "manual mutation to formate" – what does this mean?
- "minimized" – using what algorithm?
- "constrained" – how? With what approach and what force constant?
- Was a 1 fs time step really used for both equilibration and production? Why was this used vs a more conventional 2 or 4 fs time steps.
- Fig 3A doesn't add anything at all to the analysis – it's simply an image of a simulation box. Same goes for Supp Movie 1
- "custom Python script" this needs to be made available via GitHub, otherwise I can't comment on the data analysis.

Version 2:

Reviewer comments:

Reviewer #4

(Remarks to the Author)

I thank the authors for their revisions and comments. It is great that the authors have added MD data to support their experimental analysis, and I think it bolsters the study. I think (hope) that the additional sampling and analyses improves the study and strengthens their conclusions.

The changes they have made now assuage my concerns.

I make an optional suggestion based on their newly included data:

- The RMSD of the membrane in S13B isn't a particularly useful metric. There are better ways of showing membrane equilibration, but this probably isn't that necessary for the current work. I suggest this is dropped with just the protein backbone RMSD plotted.

I would also like to raise a suggestion as feedback for future studies:

- Having two copies of FocA in separate membranes is an interesting approach. However, to my mind, these should ideally be further than 2 nm apart. Waters and ions are strongly impacted by membrane so there's a risk of weird biophysical effects occurring between two close membranes. Also, dual membrane systems can be problematic with solvent densities, ion concs etc. In the current study (looking at transport in the centre of a channel) I don't consider it an issue, but for future studies I suggest sticking with a single membrane if possible.

Answer to the Reviewers' comments.

Reviewer #1 (Remarks to the Author)

Tüting and co-workers report on cryo-EM structure of the FNT family integral membrane protein FocA from Escherichia coli as wild-type protein and as a H209N variant at 2.6 Å and 3 Å resolution, respectively. Using crosslink-MS to support molecular docking, they also present a model for the interaction of the pentameric FocA with the formate producing, cytoplasmic enzyme pyruvate:formate lyase (PflB). A crystal structure of the same protein was reported by Shi and co-workers in 2009 at a resolution of 2.2 Å (3KCU), and a comparison of the experimental maps shows that the EM maps of the present work are equivalent in quality. The structural analysis is carefully conducted and sound and the models are of very high quality. The EM structure align with the earlier crystal structure with r.m.s. deviation around 0.3 Å for all atoms, meaning near identity. Furthermore, two orthologs of FocA were reported from V. cholerae (3KLY) and S. typhimurium (3Q7K).

Answer 1.1:

We thank the reviewer for their comments. The reviewer is correct in stating that the structures have near-identical parts with an r.m.s.d. of 0.3 Å; However, this is not the case for all of the atoms. Firstly, the RMSD for all atoms, calculated by PyMOL is 2.4804 ± 0.204 Å (using no atom exclusion); second, the structure resolves the N-terminal helix, which is not present in 3KCU (Wang et al., *Nature* (2009)), and therefore, entirely maps the channel; and third, this helix affects the spatial localization of the helix 90-103, which is important for formate/formic acid translocation, and which is described in the manuscript. We have carefully considered the comment of the reviewer and explicitly included in the Results section the comparison of the PyMOL-derived full-atom r.m.s.ds: *"The overall structure exhibits differences to the previously published FocA crystal structure² as the all-atom root-mean-square deviation (RMSD) is 2.5 ± 0.2 Å."*

In their analysis, the authors then assess the role of hydrophilic checkpoints around a central, conserved histidine (H209) that is essential for the unusual bifunctional role of this protein, in that it can act as an efflux channel for neutral formate, or alternatively as a secondary active proton/formate symporter for uptake. The functionalities are connected via a pH-dependent switch mechanism, and in the mentioned H209N variant, only the passive efflux functionality is retained. The authors analyse the polarity of the translocation pore found in each FocA monomer, and describe a central cavity around H209 that is separated from the two exits by a hydrophobic barrier. This feature was already found in the earlier crystal structure, and this should be mentioned more clearly in this manuscript.

Answer 1.2:

We apologize to the reviewer for not referencing the Wang et al. paper when we first introduce the channel description. We have now added the following sentence to the Results section: *"Comprehensive channel analysis (Methods) validated the described amphiphilic properties previously elucidated²."*, now appropriately referencing the work. In addition, at the final paragraph, we state it again: *"...polarity distribution, also identified previously in FNTs²⁻⁵, but further exemplified here in the full FocA channel."*

An interesting and new aspect is that in the H209N variant, the central cavity seems to have lost its polar properties, which is in line with retaining the function as an export channel for neutral formic acid, but not as an importer for formate anions. Additionally, the mutant structure contains an additional density feature in the channel that can be interpreted as bound formic acid, which at the given resolution cannot be assigned unambiguously.

Answer 1.3:

As the reviewer correctly points out, we chose not to model this density, following the Wang et al. paper where the Fc-Fo density is present at a similar position in the native protein. To address the reviewer's comment, we performed molecular dynamics simulations by including both formate and formic acid in the channel, at the resolved density coordinates, both in the native FocA protein and its H209N variant. Our MD simulations corroborate the presence of formate in the density because its residence times close to residue 209 are higher. These results are now included in the Results section *"To understand the retention...identified in crystal structures of FocA^{2,3}."* and the updated Methods sections *"Molecular Dynamics Simulations"* and *"MD data interpretation"*. The new **Figure 3** is now included, with detailed description of its four subpanels, as well as **Figure S14**, which

shows the stability of the MD simulations. Finally, the acknowledgements section was updated to include an M.Sc. student who aided in performing the MD: “We thank...performing the MD simulations.”, and Christian Tüting, who performed MD and their analysis.

For the docking model, a series of MS-crosslink restraints were obtained that serve to restrict all HADDOCK solutions correctly to the cytoplasmic face of FocA. This interaction, if real, is most likely highly transient, as not only PflB seems to be a dimer (Fig. 2C), but each monomer of this dimer would obviously only interact with one of the 5 protomers of a FocA pentamer, even partially blocking the others.

Answer 1.4:

As the reviewer correctly points out, cross-links accurately recapitulate the only interface that aligns with the *in vivo* localization of both proteins. XL-MS has been used successfully for more than 20 years to obtain useful and specific constraints on protein assemblies, also for transient protein interactions. The problem of obtaining false-positive cross-links was carefully addressed in our work as all MS/MS spectra were thoroughly checked and cross-links appearing only once were excluded. Non-specific cross-linking is not an issue here as the respective cross-links will fall below the sensitivity threshold of mass spectrometry and will therefore not be detected. As for the *in vivo* formation of the PflB-FocA complex, in-cell structural biology analysis would be needed, which is outside the scope of the current manuscript, but is considered for future studies and is mentioned in the revised version of the manuscript. We have added the following sentences in the Results section: “*The proposed model underlines...tomography would be relevant*”. Finally, Jana Lorenz, who performed and evaluated cross-linking experiments, was moved from the acknowledgements to the author list as her contribution was re-evaluated to be sufficient for co-authorship. Respective “Contributions” section was also updated to reflect that.

On the structural side, a major issue that seems to be missing from this work concerns the structural flexibility of the periplasmic side of FocA proteins. From the first structures on, different conformations for the cytoplasmic funnel were reported, and the S. typhimurium structure then even identified different conformations for the N-terminal helix of the protein that would clearly affect conductivity through the individual pores. Earlier work has put quite some emphasis on these features, and it is notable that they are not mentioned here at all.

Answer 1.5:

The reviewer mentions two points: (a) analysis of the periplasmic side of FocA, and (b) the N-terminal helix flexible conformations that would affect conductivity. For both points, we have made revisions.

In response to part (a) of the reviewer’s comment, specifically, we analyzed the 3D variability data by also focusing on the periplasmic side. Our new results show that the protruded α -helix S228-S246 undergoes disorder-to-order transitions, captured by our cryo-EM data. Although this part of FocA is not involved in the interaction with PflB, which is the focus of the paper, it can indeed affect gating of formic acid/formate. The new **Figure 4A** includes the described analysis, and relevant text is now presented in the Results section “*3D variability analysis... and therefore, regulating translocation*²²”.

In response to part (b) of the comment, we systematically analyzed the available structures of FNT channels and focused on the conformational changes of the N-terminal helix and the structural variability when it is disordered. We also updated previous **Figure 2A**, which is now **Figure 4A** (with updated Figure legend), to globally show the flexibility of the molecule. We also added new **Figure S13** to the supplementary, referencing previous studies that have discussed this issue. Our revised text is now included in the Results section: “*3D variability analysis of FocA... ground-state, physiological pH structure of FocA.*”

Instead, the authors now conducted a 3D variability analysis (Fig. 2A) which highlighted flexibility in the C-termini, but not the N-termini. This is not in line with the earlier analysis and might be of interest. It should absolutely be mentioned and discussed in this work.

Answer 1.6:

We apologize for the confusion that might arise from a short and compactly written paper, which was aimed originally to be published as a short communication. Previous Fig. 2A pointed to the N-terminal and C-terminal helical interactions at the cytoplasmic side, and to their flexibility. We now made the 3D variability analysis a separate Figure to improve clarity (**Figure 4**). The new Figure shows the flexibility of the N-terminal helix, whereas the short C-terminal helix is only slightly

affected. This is perfectly in line with the previous results mentioned by the reviewer. We have now cited the appropriate papers in this context. We extended the text to include the earlier analysis of the conformational changes localized at that region for resolved FNT channels; please see **Answer 1.5** for exact changes in the revised version.

Related to this, the reported CryoEM structures are refined with C5 symmetry imposed, which averages out any conformational differences between monomers. It would be essential to analyse a careful refinement of this data with C1 symmetry for whether structural asymmetries within one pentamer can be identified. Note that such asymmetry on the cytoplasmic side could possibly be well in line with the symmetry break imposed by the binding of PflB and may even be prerequisite for it. Refining such rather small differences in an otherwise largely symmetric pentamer will likely be challenging, but seems warranted given the functional implications.

Answer 1.7:

This is a very critical and important comment of the reviewer. Indeed, by doing the 3D variability analysis on the symmetric pentamer, we observe conformational changes mostly on the cytoplasmically-oriented side of the protein. The question whether these are concerted or not has direct implications for various functions of the channel. To address this comment, we have performed: (a), an analysis of the C1 reconstructed cryo-EM map; and (b), an analysis of the molecular dynamics simulations, which were purposely run for the fully free pentamer (without symmetry operations).

C1 reconstruction was performed and attained a resolution of 2.87 Å. All data are deposited in EMDB and PDB, and are also included in **Table 1**. Local resolution analysis clearly showed distinct regions across the protomers with different resolvability, corroborating the asymmetric nature of FocA. This analysis is now included in the new **Figure 4** as subpanel B, substantially extending our Results section. The modelled asymmetric structure shows conformational fluctuations of residues localized on both the cytoplasmic and the periplasmic sides of FocA, but not in the membrane or inner part of the channel, corroborating the localized asymmetry hypothesized by the reviewer and predicted by our docking model. We now revised the Results section and added the following text: *“To identify if the FocA pentamer...the 3D variability analysis (Fig. 4A).”*

MD simulations demonstrate that the pentamer can acquire many more conformations than those modelled by our cryo-EM data. Indeed, inter-protomer calculations of RMSDs show a distribution of values up to 10 Å. These results are shown in **Figure S14**. The Results section is now extended to include these new calculations, text: *“These regions were independently...interactions via conformational selection mechanisms²⁶.”* The Methods section was also updated to reflect the C1 reconstruction calculations, as well as **Figure S2**, to add the reconstruction workflow.

Not unexpectedly, a quick run of AlphaFold3 also showed that all solutions, while close to the experiment, are fully symmetrical and do not reflect asymmetric interactions. Interestingly, including PflB in these predictions also consistently places the enzyme close to the cytoplasmic face of FocA, but at a certain distance and not in a consistent orientation.

Answer 1.8:

We highly appreciate the reviewer's review of our paper and their comments. We have made similar observations with AlphaFold3, which tries to enforce symmetry and protomer-specific variability becomes barely visible. We have observed this not only for FocA, but also for other structural projects in our lab. We think this is due to computational limitations of the underlying hardware, but this really limits the usefulness of AlphaFold in the analysis of structural flexibility and local fluctuations and, in particular, warrants caution in its use and the conclusions that may be drawn. Nevertheless, based on the reviewer's comment, we also ran AlphaFold3 with the indicated 7 polypeptides (5x FocA, 2x PflB). This analysis revealed some very interesting results, which we would like to highlight here:

- 1) The N-terminal helix and its immediate neighborhood exhibit clearly lower pLDDT scoring, indicating a lower per-residue confidence. This is perfectly in line with the observed flexibility in our cryo-EM data (see **Figure 4**) and agrees with our previous observation for a correlation of lower, localized pLDDT scores with function, e.g., as we demonstrated for the endogenous myo-inositol-1-phosphate synthase and all structurally-resolved isomerases (DOI: 10.1073/pnas.2400912121).
- 2) The dimeric PflB and the pentameric FocA structures appear to be properly predicted in their unbound states.
- 3) The PAE scoring matrix clearly shows that AlphaFold3 predicts that FocA and PflB are **not** interacting. The error between the dimeric PflB and the pentameric FocA is at the upper limit of 31

Å, whereas the PAE between the protomers of PflB and FocA shows a low, to near zero, PAE score, respectively.

The latter is the most important scoring matrix when analyzing AlphaFold3 multimeric structures. The prediction window, which appears to be a function of total residues, is a spherical simulation shell. AlphaFold3 places all atoms within this box and follows the paradigm “predict everything the user prompted”. This leads to artificial protein-protein-interfaces in the atomic coordinates, and even knotting/clashing/etc. of long disordered regions. We recently published a short Perspective article, discussing this (Träger et al., *Structure*, (2024)). To conclude, the Alphafold model of the complex is not informative because of low confidence in the PAE scores.

In summary, this short manuscript reports on a technically sound re-determination of an earlier crystal structure by cryo-EM. It is well written and interesting, and adds valuable information where in the crystal structure protein termini were disordered, and otherwise largely confirms the state of knowledge. Terms such as 'polarity hub' describe findings that were already made in the earlier works and it is not clear to this reviewer that such re-branding is helpful. The issue of symmetry breaks, both within the FocA pentamer and in its predicted interaction with PflB should be elaborated on, and for this the docking experiment could possibly be expanded to the available unsymmetrical structure of V. cholerae and S. typhimurium.

Answer 1.9:

We thank the reviewer for their positive assessment of our work. To address their final summarizing comment, we have recalculated the asymmetric cryo-EM map and its model structure (see **Answer 1.7**), and elaborated more on the asymmetry of FocA. Furthermore, we have performed molecular dynamics simulations and additionally expanded on the asymmetric nature of the protein (see **Answer 1.3**). The suggested docking experiments for *V. cholerae* and *S. typhimurium*, without cross-linking data is an exercise that will not allow conclusions to be drawn regarding similarities or differences in the binding modes, as cross-links are crucial for model generation. Overproducing, purifying, cross-linking, and modelling FocA- and PflB-specific proteins for both organisms is out of the current scope, as the manuscript resolves and focuses on the interface of *E. coli* FocA-PflB, and is not feasible in the current time-frame. Finally, we have revised the term “polarity hub” to follow the reported “amphipathic channel” and “core”, which is now used throughout the manuscript.

Reviewer #2 (Remarks to the Author)

In this manuscript, Tüting et al. present a comprehensive study on the structural and functional relationship of FocA from E. coli, a member of the formate-nitrite transporter (FNT) family that mediates formic acid transport. The authors utilized cryo-EM to resolve both the wild-type and H209 mutant forms of FocA. Additionally, they employed cross-linking mass spectrometry (XL-MS) to map the interactions between FocA and the cytosolic enzyme PflB, subsequently generating a 3D model based on this data. The structural analysis provided in the manuscript is robust and offers valuable insights into the mechanism of this important FNT family. However, the proposed mechanism could benefit from more direct functional studies to strengthen the conclusions. The revised version of this manuscript would be a strong candidate for Nature Communications.

Answer 2.1:

We thank the reviewer for this brief and positive summary of our submitted manuscript. We have performed extensive revisions, correlating our data and results with the available literature, and extending our functional insights into FocA structure and function with two additional major revisions: (a) we have performed molecular dynamics simulations to understand the translocation of formate/formic acid and the role of H209 and its H209N variant. This analysis has derived entirely new insights into the mechanism of translocation of the substrate (see **Answer 1.3**); and (b) we have analyzed the asymmetric nature of the FocA channel, performing additional reconstruction of its C1 structure, which points to local flexibility for each protomer (see **Answer 1.7**).

I have the following suggestions and questions for the authors:

Major:

1. The finding that the H209N mutation converts a bidirectional anion transporter into a unidirectional one is intriguing. However, these results seem to contradict transport assays from Eric's lab, which used radiolabeled substrates and demonstrated that the H209N mutation

abolishes transport (<https://www.embopress.org/doi/full/10.15252/emj.201695776>). Could the authors provide comments on this?

Answer 2.2:

Previous *in vitro* studies and *in vivo* transport analyses performed in a heterologous yeast system were well-conducted and at the time yielded new information regarding how these proteins might function. Meanwhile, the possibility of adopting a homologous *in vivo* system using *E. coli*, the natural host of FocA, has revealed that control of formic acid translocation is considerably more complex than initially thought. This is presumably also the case for other members of the FNT family. The *in vivo* findings we report here and in recent publications (e.g., Kammel et al., *Microbiology (Reading)*, (2022); Kammel and Sawers, *Microbiology (Reading)*, (2022)) and the previously obtained *in vitro* results cannot be compared directly. Due to the fact that the enzyme pyruvate formate-lyase (PflB), which gates *E. coli* FocA *in vivo*, and the formate hydrogenlyase (FHL) complex, whose activity is coupled to formate uptake by FocA, are essential for FocA to conduct its *in vivo* functionality, this explains how FocA works in bi-directional formate/formic acid translocation in the homologous *in vivo* system and why the previously published very nice data from the Beitz group must now be updated and partially reinterpreted. The previously published findings are not ‘wrong’, but represent the status of the field at the time of publication. The new experimental set-up we are able to use has revealed the role of interaction partners and other regulatory features not apparent *in vitro* or in a heterologous expression system. *In vitro*, the H209N variant of FocA is non-functional; however, *in vivo*, the variant is functional exclusively as an exceptionally efficient formic acid efflux channel (DOI: 10.1099/mic.0.001132), which is also shown in **Figure S11**. The H209N FocA variant does not translocate formate into the cell *in vivo*, which is actually in agreement with the lack of uptake of formate in the heterologous system, but the reasons for this are because the histidine residue within the pore is essential for pH-dependent uptake of formate. This point has now been carefully expanded upon in the revised text: “*Past in vitro and in vivo studies...pH-dependent formate uptake.*”.

2. Line 34, “The FocA-H209N efflux-only variant reveals a density consistent with formic acid located directly at N209, breaking local polarity distribution,” suggests that formic acid is bound to FocA-H209N. If this substrate is from a native source, it might be worthwhile to use LC-MS or native mass spectrometry to confirm the identity of the substrate bound to FocA-H209N.

Answer 2.3:

We thank the reviewer for this comment. We also discussed potential experiments to identify this density. Native mass spectrometry is unfortunately not possible, as the FocA pentamer is solubilized with an undefined number of DDM detergent molecules, thereby a broad distribution is to be expected and the mass-resolving power of the mass spectrometer used for native MS measurements limits detecting such a molecule. Additionally, formic acid is used to acidify LC/MS running buffers, so a high formic acid background further prevents such an analysis. Direct structural methods (extensive cryo-EM at a facility hosting a 300 kV with energy filter) might also not be able to resolve this question, as we expect this molecule to be intrinsically flexible in its spatial orientation, and the resulting density might still be ambiguous. Notably, similar densities, which were not modelled, were also identified near H209 by Wang et al., 2009 for *E. coli* FocA, and for FocA from *Vibrio cholera* by Waight et al. 2020 after crystal-soaking with high concentrations of formate. However, to satisfy the reviewers’ comment we have addressed the likelihood of formate or formic acid presence in the channel with molecular dynamics simulations, please see **Answer 1.3** for details.

3. The functional assays were conducted using whole-cell systems. To provide more direct evidence supporting the proposed mechanism, such as unidirectional transport in the H209N-FocA mutant, the authors might consider using reconstituted proteoliposome-based assays.

Answer 2.4:

We thank the reviewer for suggesting alternative methods for the investigation of *in vitro* translocation activity. Based on the detailed answer given above to the comment of the reviewer (**Answer 2.2**), inclusion within proteoliposomes of FocA is currently outside the scope of this study, particularly as the focus of the study was more a structural one. Nevertheless, in an attempt to address this issue, this is why we chose instead to perform the molecular dynamics simulations, which also addresses the structural aspect of the cryo-EM and the *in vivo* findings. Future *in vitro* analyses with proteoliposomes are planned and will be attempted with isolated FocA, formate

hydrogenlyase complex and PflB. However, this will likely be quite time-consuming and will pose a major technical challenge.

4. How does the H209N mutation cause FocA to become a unidirectional efflux transporter? Can authors explain this mechanism using the structural information they have obtained?

Answer 2.5:

We thank the reviewer for suggesting to interpret our structural data in the light of the *in vivo* measurements reported in **Figure S11**. In order to address this question, we have performed molecular dynamics simulations utilizing different protonation states of H209, as well as simulating the H209N variant. Translocation substrates used were both formate and formic acid (see **Methods**, sections: “*Molecular dynamics simulations*” and “*MD data interpretation*”). The results show that H209 can retain both formate and formic acid in its proximity, thereby pointing to a protonation mechanism for translocation. H209 tautomers with partial charges can delocalize preferentially the formate (new **Figure 3**). Therefore, our MD analysis provides a biophysical explanation for the role of the H209 residue in translocation. This was a prerequisite to analyze the H209N variant.

MD analysis of the H209N variant shows that formate changes localization towards the periplasmic side of the channel and not towards its cytoplasmic side (**Figure 3D**). To further address this comment, we also analyzed the efflux data from Figure 4 and new influx simulations by placing the substrates at the periplasmic side of residue 209. Respective plots are now included in new **Figure S14C**. Panel **C** specifically shows the probability (out of 1) for the substrate to change localization via residue 209 towards the periplasmic (efflux) or the cytoplasmic (influx) side of the channel.

These plots show that:

- (a) Efflux is possible for nearly all simulated systems (two plots on the left – H209N variant; all H209 protonation states), while only fully protonated, positively charged His and the negatively charged formate does not allow translocation in the efflux direction (first plot on the left, HSE and HSP-related plots).
- (b) Influx is only possible for the fully protonated histidine (two plots on the right), while for H209N only neutral formic acid can translocate at a very low rate. It is unsure if this state is relevant, or even physiologically possible, but the main conclusion is that for H209N essentially only efflux is observed, while His protonation regulates both influx and efflux.

To conclude, we performed and analyzed extensive molecular dynamics simulations based on our cryo-EM data, and derived mechanistic insights, which are now included in the Results section: “*To understand the retention of formate...densities identified in crystal structures of FocA^{2,3}*.”, and the two new Figures mentioned above.

5. Including a summary diagram of the proposed mechanism would be highly beneficial for readers.

Answer 2.6:

We thank the reviewer for proposing to include a summary diagram. We have explained in **Answer 2.5** a mechanism supported by the identified density in the mutant and the molecular dynamics simulations data (**Figure 3**, **Figure S14**). To propose a model for translocation, NMR measurements would be needed to deconvolute the protonation states H209, which is very challenging to be performed for a homomultimeric membrane complex. Moreover, the focus of the paper is on the asymmetric nature of FocA and its binding to PflB, as well as the definition of the residues of the entire channel. We have indeed elaborated on these mechanistic insights showing the inherent asymmetry of FocA utilizing molecular dynamics simulations, asymmetric reconstructions of our cryo-EM data, and the docking model, which, we feel, are sufficient to support the asymmetric nature of the channel.

6. In the pentameric FocA transporter structure solved in this study, structural variability is observed. Is it possible that the protomers may move independently of each other? To better address potential conformational heterogeneity, could the authors clarify if they have attempted to perform C1 symmetry to reduce symmetry restrictions or applied symmetry expansion on the final map (C5) followed by focused classification on a single protomer?

Answer 2.7:

We thank the reviewer for this comment. During the revision process, we have performed asymmetric 3D reconstruction of FocA, and with local resolution calculations have observed asymmetric features in each protomer. In addition, we have analyzed the MD data and show that

the whole structure moves asymmetrically, with RMSDs reaching 10 Å across protomers. For details, and changes in the revised manuscript, please see **Answer 1.3**.

7. Rotamer outliers (table 1) 11.26% and 2.26% in these given resolutions are too high. Authors need to validate their coordinate models.

Answer 2.8:

We thank the reviewer for pointing out this issue. We re-refined the atomic models in their respective EM densities, and updated the PDB depositions. For the native FocA we have now 1.8 % and for the H209N variant 0.90 % rotameric outliers. The remaining outliers could not be further resolved, as none of the library-defined rotamers (using COOT) is in agreement with the EM map, and we believe that the experimental EM density must be favored. The clash-score for both increased slightly, but we believe the changes made have improved our submitted model. We invite the reviewer to inspect the maps and models. In addition, we carefully refined the asymmetric structure in the cryo-EM map, with comparable structure validation statistics. All those updated values are included in **Table 1**. Data availability section has also been updated.

8. Authors need to include Western blots to indicate the expression levels of FocA and PflB variants, ensuring that observed changes are not due to alterations in protein expression levels across all functional assays.

Answer 2.9:

We thank the reviewer for this comment. Western blot data have been published for plasmid-based expression of the mutated *focA* gene encoding FocA-H209N and these data show that this protein is very stable and synthesized at a level similar to that of native FocA when cells are grown under the same conditions (please see Kammel et al, *Microbiology (Reading)*, Jan 2022, Fig. S1). As the *in vivo* efflux data in the current manuscript were determined using an *E. coli* strain where the mutation in *focA* to change codon 209 (encoding FocA-H209N) was introduced in single copy on the chromosome, replacing the native *focA* gene, a western blot of this single-copy FocA protein is difficult, due to sensitivity limits of the antibody against FocA. However, as the plasmid-based synthesis levels and stability are robust and like wild-type, we assume that the same holds for single-copy situation.

Regarding PflB, we present in the revised manuscript a western blot (new **Figure S12**) in which we show that PflB is stably synthesized in the mutant strain DH4200 (encoding FocA-H209N). We have also added the following text in the Results section to address the comment of the reviewer: “*Immunological analysis of plasmid-encoded...effect on FocA’s interaction partner, PflB.*”. In addition, related Methods section has been slightly revised. Taken together, these data indicates that the variant FocA is produced at similar levels to the native, wild-type FocA protein in the *in vivo* analysis, and that PflB is also stably synthesized in these cells.

9. The assembly of the FocA-PflB complex is based on a 1:1 model. Do the authors have any evidence supporting this stoichiometry, such as data from size-exclusion chromatography (SEC)?

Answer 2.10:

Since we have so far been unsuccessful in purifying a stable FocA-PflB complex (including attempts with size exclusion chromatography) over the last 10 years, we have not performed a size exclusion chromatography for the current study, as was suggested by the reviewer. Due to the apparent transient and unstable nature of the interaction and, particularly because of the necessity of using a detergent to solubilize FocA and maintain it in a ‘soluble’ state in aqueous buffer systems, we have noted that this further destabilizes the interaction considerably. We hope with future advances in in-cell cross-linking and whole-cell cryo-electron tomography that such issues can be addressed more directly. To address the comment, have included in our manuscript the following: “*The proposed model...cryo-electron tomography would be relevant.*”

10. FocA contains five channels for anion transport. In the FocA-PflB complex model, only one channel appears to be channeling, which could reduce the efficiency of the transport process.

Can the authors provide insights on this?

Answer 2.11:

We thank the reviewer for this important comment. We expect formate either to be directly shuttled or diffusion-controlled. If directly shuttled, one or two channels might be transiently interacting with PflB, while others, for a very brief amount of time, might be unavailable for translocation. However, the docking sites are increased and are presumably highly flexible with PflB possibly moving from

N-terminal helix to the next very rapidly, so the interaction is more likely 2 from PflB and 5 from FocA. If formate entry is diffusion-controlled, then all sites are available, but at the risk of acidification of the cytosol and the encountering of more unsuccessful entrance events into the channel due to the diffusion process. We have added this discussion in a short section in the revised manuscript: "*Such a structure underlines...are hindered or less efficient.*"

Minor:

1. Line 67. "*endogenous E. coli FocA channel ...*" The term "*endogenous*" typically refers to protein expression from genomic DNA with a native promoter. If this is the case, the authors are encouraged to provide a more detailed description in the Methods section.

Answer 2.12:

Thank you, this has been revised.

2. Line 315, "*FocA pore analysis and derivation of the integrative FocA-pflB complex with*". The text should be "*FocA-PflB*" instead of "*FocA-pflB*."

Answer 2.13:

We thank the reviewer for finding this typo. We have now corrected this.

3. *The raw mass spectrometry data should be deposited to the PRIDE database.*

Answer 2.14:

We have uploaded the mass spectrometry data to 'Pride' and updated the Data Availability section accordingly. The reviewer-access is:

Project accession: PXD054538

Project DOI: Not applicable

Username: reviewer_pxd054538@ebi.ac.uk

Password: H5EN3SMNgaxX

4. *Please consider labeling additional side chains around the density in Figure 1H.*

Answer 2.15:

In the interest of clarity and simplicity, we would like to keep the focus in this panel on the bound density, thereby not labelling any additional side chains. We have, however, made a separate Figure (now **Figure 2**) to improve clarity.

Reviewer #3 (Remarks to the Author):

In this manuscript Tuting, Janson and Kammel et al describe the structural characterisation of FocA, a formate nitrate transporter. This manuscript is overall well written, but is very concise – maybe too concise in places. I would recommend adding some additional details at the point of resubmission. There is potential for re-ordering and for moving figures from the supplementary information into the main text to improve the clarity of the manuscript.

Answer 3.1:

We thank the Reviewer for this positive assessment of our manuscript. The manuscript was a direct transfer from NSMB, and it was written as a brief communication, which is why it was so concise. We have addressed the reviewer's comment by substantially expanding on the text, including also additional references (N=15) that appropriately put our results and discussion into context. We then made 5 Figures instead of 2, and separating the descriptions of the Figures to make these and the accompanying argumentation easier for the reader to follow. Additionally, during the revision process, we have further strengthened the molecular insights of our manuscript by performing molecular dynamics simulations (see **Answer 1.3**) and asymmetric reconstructions of FocA to derive localized flexibility (see **Answers 1.7** and **2.7**). We also slightly revised the **Methods** section. Now, **Figure 1** includes an overall description of FocA; **Figure 2** shows the molecular insights derived from the cryo-EM structure, focusing on the channel; **Figure 3** describes the MD results, which highlight the role of protonation of the histidine residue, as well as of the Asn variant; **Figure 4** shows the flexibility of FocA, both utilizing symmetry and asymmetry in its reconstruction; and, finally, **Figure 5**, shows the model for its interaction with PflB.

My expertise lies in the field of structural proteomics and membrane protein biology, so I do not feel able to comment on the quality of the cryoEM. These experiments are well performed, but the lack of clarity and detail in this manuscript makes it hugely difficult for the reader. I recommend

that the authors should revise their manuscript keeping this in mind. I only raise here minor comments, which I'm sure can be addressed simply.

Answer 3.2:

We thank the Reviewer for providing us with feedback on the structural aspect of the manuscript. We have revised the paper and explained better the variability analysis, which was too dense for the reader. In addition, revised text is now put into context, correlating better with the available literature. As this is a general comment from the reviewer, we point to the highlighted parts of the manuscript.

Minor comments:

-line 34, need to define efflux-only variant

Answer 3.3:

We have revised this part, and explained it in the Abstract: "A *FocA-H209N* variant that exhibits an efflux-only channel function in vivo".

-line 35, what do you mean by 'breaking local polarity distribution'?

Answer 3.4:

We have revised the term to align with the literature and it is now mentioned as "amphiphilicity".

-line 57. The authors refer to Fig S1b where a 10mer is observed by BN-PAGE. It would be useful for the authors to comment here the biological relevance of this higher order assembly, and the precedent (if any) for their formation from previous work.

Answer 3.5:

This decameric form has been observed many times, originally in purification studies performed by Hunger et al., *Biol Chem*, (2014) and Hunger et al., *Front Microbiol*, (2017). The pentamer is in the membrane and therefore cannot form dimers of pentamers, except if there is a vertical, side-by-side interaction of FocA. However, this is not observed in cryo-EM, as e.g., in photosystem dimers. Therefore, the decamer is most likely a non-physiological state. We added this comment within the text, below Figure 1A: "Potential decameric forms arise during membrane solubilization and purification³²."

Figure S4 could be moved to the main text.

Answer 3.6:

While we understand why the reviewer made this suggestion, as this data represent an analysis of the quality of the new model and do not hold any substantial novelty, we prefer to keep it in the supplementary. We have followed the reviewer's general suggestion to make our manuscript more accessible to the general audience, please see **Answer 3.1**.

-Line 71 – what do the authors mean by 'flattest surface'. The authors should clarify this in the text. Once I see the figure it makes sense, but it would be nicer to see this in the text.

Answer 3.7:

We thank the reviewer for the comment. We added the explanation to the text: "This flatness of the surface is imposed by the entirely flat positioning of the N-terminal α -helix on the cytosolic face of the FocA pentamer, as compared to other FNTs (Fig. S5, Fig. S6)."

-The authors 'trace the formate channel' in Figure 1E. It is not clear to me however where this fits in the context of the full length protein oligomer. An additional figure, or figure inset may help to orient the reader.

Answer 3.8:

We have added a subpanel explaining where in the structure the channel is in the context of the pentameric, full-length FocA, it is updated **Figure 1D**.

- line 83 – 'Comprehensive channel analysis' what do the authors mean by this?

Answer 3.9:

We thank the reviewer for their comment. Indeed, it was not entirely clear. We added now a reference to our Methods section, where this is described. In detail, our channel analysis includes an investigation of the volumetric dimensions of the path, calculated by the radius (including side chains) and free radius (only considering backbone atoms), a mapping of all involved residues, and an analysis of the polarity of the channel. Additionally, major properties of the FocA channel,

including sequence conservation and important features (e.g., the Phe-gate), and additional EM densities are identified, not only for the wild-type FocA, but also for the H209N variant. The term “comprehensive channel analysis” is, thus, used to indicate the breadth of this analysis which is now better described in the Methods section under “*Channel analysis and sequence alignments*”.

- *Figure 2A – really unclear to see the motions the authors are referring to here.*

Answer 3.10:

We thank the reviewer for their comment, we have updated Figure 2A (now Figure 4A) to make the motions clearer. Motions are also shown in context, i.e., for both cytoplasmic and periplasmic face.

- *line 108 – the authors derive an integrative model for FocA-PfIB. Why did they approach this challenge in this way? Why not an additional cryoEM structural analysis?*

Answer 3.11:

Overall, purifying the two proteins together is a significant challenge as the complex is highly transient and unstable, it includes a membrane partner (requiring detergent) and a soluble partner, and such work has not produced any fruitful results after several years of attempts. For details, please see the detailed **Answer 2.10** above.

Fig S12 – needs more explanation how these data guided the modelling. What do they show? For the non-expert reader this needs much more clarity.

Answer 3.12:

The reviewer is correct and we have now added detailed information in the Figure legend, providing more explanation about the data in B and C and how their interpretation. An extension to the Figure legend has been added, now being legend of Figure S16, after including new Figures during revisions. These data were not used to drive the modelling procedure, but as data that validate the importance of the N-terminal helix of FocA for interaction with PfIB and formate transport. Finally, we have updated the manuscript text in the Results section, stating: “*Mutagenesis studies, in which...legend details the experimental strategy.*” We would like to thank the reviewer for this suggestion.

- *Figure S13 – the authors state in the text that the structure of PfIB bound to a substrate was used for integrative modelling. Why is there an AlphaFold2 model shown in Figure S13A? How was this used?*

Answer 3.13:

We thank the reviewer for their question. For docking, we used the dimeric PfIB AlphaFold2 model, in which the translocation substrates were superimposed. We avoided using a crystal structure as there are 8 structures already and one shows a major interface change (1QHM), making it ambiguous for docking selection.

- *Line 117 – what is their new ‘symmetry agnostic clustering algorithm’. Whilst the methods goes into a lot of technical detail about this, a simple few sentences explaining this would be beneficial.*

Answer 3.14:

We have edited the specific Results part, adding a description of the algorithm as requested by the reviewer: “During this procedure, all possible rotations across symmetry points of FocA are sampled, and then the lowest RMSD complex is used as reference for subsequent clustering. This procedure overcomes the limitations of chain-ID-dependent clustering implemented in HADDOCK.”

- *Why do the authors crosslink between the FocA N-terminal peptide and PfIB? Why were so few inter-protein crosslinks detected for the full length protein? Did the model require all crosslinks, including those from the peptide to reach a satisfactory confidence score?*

Answer 3.15:

Revised **Figure S16** shows that the N-terminus of FocA is important for PfIB function (see **Answer 3.13** above). This is why the peptide was used for docking. The inter-protein cross-links for full-length proteins are low, possibly due to the transient nature of the interaction. All cross-links were used for docking, but random removal of restraints is inherent in the HADDOCK software. Therefore, by default, 50% of cross-links were randomly removed to generate each docking solution. Satisfactory solutions are reached by having converging clustering (see **Figure S17**).

Reviewer #1 (Remarks to the Author):

In their revision, the authors have extensively and thoroughly addressed the issues raised by this reviewer, and apparently also by the others. This is a very conclusive and informative study of FocA and its interaction with the enzyme PflB. I fully support publication in Nature Communications and have no further request or criticism. Congratulations to the authors.

Answer 1.1:

We thank the reviewer for their positive assessment of our revised manuscript.

Reviewer #2 (Remarks to the Author):

I consider the revised manuscript, 'Conserved Hydrophilic Checkpoints Tune FocA-Mediated Formate:H⁺ Symport,' suitable for publication. The authors have thoroughly addressed the reviewers' concerns with thoughtful responses and necessary revisions. Including MD simulations and reprocessed data enhances the mechanistic insights, significantly strengthening the study, particularly regarding the substrate translocation mechanism.

Answer 2.1:

We thank this reviewer for their positive assessment of our revised work and we are happy that they are convinced with our mechanistic study on FocA.

Reviewer #3 (Remarks to the Author):

The authors have done a thorough job at addressing all my comments and I have no further issues to raise.

Answer 3.1:

We thank this reviewer for their assessment and support for acceptance of our manuscript.

Reviewer #4 (Remarks to the Author):

This is a revision of a previously submitted manuscript, and I have been asked to assess the quality of the MD simulations. These have been added to address the EM density, and predict whether it is formate or formic acid.

Answer 4.1:

We are grateful to have our manuscript also reviewed by an MD expert, as we believe that this further strengthens our manuscript, as these simulations were performed to augment the major findings from the *in vivo*, cryo-EM and the cross-linking MS experiments. We want to point out that none of the previously reported results is contradicted by our, now adjusted, extended analysis, which basically expands the previous **Fig. 3D**. On the other hand, the manuscript has now been revised to display in detail the interaction preferences of the substrates with the FocA channel with simulations of 200 ns in triplicates per condition. Below we answer the concerns of the reviewer.

Question 4.1:

I have some major concerns about the MD data. These are as follow:

The simulations are very short – 20 ns. And from what I can see no repeats were made for each condition. If this is the case, then there is a chance that the simulations are severely undersampled. I would normally expect at least 200 ns and at least 3 repeats. Ideally 5-10. If there are repeats this should be made clear in the methods, and their densities should be plotted separately to demonstrate the degree of agreement between them.

Answer 4.2:

We fully agree with the reviewer's concerns about the simulation length and therefore extended this part of our work. The following measures were taken to address this concern:

- 1) We optimized the simulation box. Instead of having a single membrane within the simulation box, we placed two facing membranes with bound FocA within a single simulation. The distance between the membranes is long enough to avoid long-distance disturbance ($>20 \text{ \AA}$). This was performed to increase sampling, as the reviewer suggested and is now presented in the new version of **Fig. S13**. This is mentioned in Results: "The set-up included ..." and **Methods** sections "Two membrane systems...long-distance interactions." and "Additionally, the Ca atom ... channel observations."
- 2) Simulations were run for 200 ns with a timestep of 2 fs. Note that all conditions, namely amino acid variations (HSD, HSE, HSP and ASN) and substrate protonation state (FORA and FORH), were run in triplicate.
- 3) Revised Methods and Results parts, to unambiguously describe our workflow and derived data, including the agreement between independent simulations. Specifically, regarding this point, the reviewer can see the RMSDs in the new version of **Fig. S13B**.

Question 4.2:

The analysis presented in 3D is a distribution of all positions of the formate/formic acid from the MD data. To understand the data better, we need to see the time series data too – this will let us know if the simulations have reached any kind of equilibrium or are still converging. Given the very short timescales, the former seems very plausible. I believe that this is a requirement for publishing in Nat Comms anyway.

Answer 4.2:

We performed the MD for 200 ns in replicates as suggested by the reviewer (please see **Answer 4.1** above). We have now revised **Fig. 3** according to the reviewer's comment. **Fig. 3** shows the time that formate spends in and out of the channel (binned in 8 categories). **Fig.**

S13 shows this for formic acid. As the reviewer can see, we observe the substrate across conditions in all different positions, but of course with preferences. All these plots show that the substrates across the simulations are properly sampled in respect to their diffusion within the channel. Based on our cryo-EM data, we were able to define the initial localization, which reduced the degrees of freedom of our system.

Question 4.3:

The formate/formic acid were placed in the EM density. How? How confident of the orientation of the molecule are we? Would this be the same for the different charge states? To address this, I would have personally performed an extensive (100 ns) equilibration where the molecule could rotate but not shift in the Z dimension. But in absence of this, the protocol should be more clearly described as at the moment it is hard to determine what exactly has happened.

Answer 4.3:

We thank the reviewer for their comment on our integrative paper, which combines biochemistry, cross-linking MS and cryo-EM with MD analyses, the area in which the reviewer has expertise. Placement is based on the derived cryo-EM-density map and was fitted with the ChimeraX function “fit in map”. This function “locally optimizes the fit of atomic coordinates into a density map” (ChimeraX manual). This is standard in the cryo-EM field. We added the following text to the Methods: “The substrate was positioned near residue 209 at Z=-3.14 for the first membrane and Z=-90.2. This placement is derived from the cryo-EM map, where the molecule was rigid-body fitted using ChimeraX”. Additionally, during revisions, we added a 5 ns equilibration step, in which we fixed the substrate in space, by restraining the carbon atom by 5 kcal*mol⁻¹*Å⁻², but which allowed for free rotation. This change ensured that during our all-atom simulation, we were not biased by the initial orientation. This is now also mentioned in the Methods: “In all equilibration steps...Particle Mesh Ewald (PME).”

Question 4.4:

The orientation of the formate/formic acid over time should be analysed. As should the contacts it makes with the channel.

Answer 4.4:

We thank the reviewer for this comment. The reviewer’s suggestion about contact analysis was indeed excellent: Our new contact analysis of the 200 ns triplicates uncovered specific residues where the substrates prefer to localize. This is now presented in the new **Fig. S16**. Most of these preferential contacts of the substrates are with, or in proximity to, residue 209, further corroborating that the EM density we discovered could correspond to a bound substrate. For the other comment, as the substrates are mesomeric molecules, orientational analysis will be extremely challenging, requiring constant-pH MD, QM/MM, or other methods, which are not directly related to the main message of our paper and which focuses on the cryo-EM structure of FocA, its efflux-only variant, and its higher-order interactions with PflB.

Question 4.5:

What about x/y dynamics? In general the formate/formic acid motions seem confusing to understand. This is something that a video would be useful to show.

Answer 4.5:

Each FocA has 5 channels through which formate can pass (it does not pass through the central pore, which is filled with lipids). These 5 channels are very narrow and do not permit free X-Y movement. For example, **Fig. 2** shows the channel, being almost exactly as the size of the substrate. A movie, in our point of view, will not add any additional insight to the

manuscript, but we made one for the reviewer, that can be accessed in YouTube here: <https://youtu.be/nAJq8bC5odw>. The video shows a protomer with formate, the viewer sees the pore from the cytosolic side (aligned perpendicularly on the XY plane). As the reviewer can appreciate, while being in the channel, the substrate only undergoes minor XY movement, and, thereby, only the Z coordinate is sufficient to locate the substrate within the channel. The movie was generated by ChimeraX morphing of every 25th frame.

Question 4.6:

In 3D, I'm not too sure about the data:

- Firstly, in all cases the molecule shifts away from the EM density, typically by at least 4 Å. Arguably this suggests that the EM density corresponds to neither formic acid or formate.

Answer 4.6:

We thank the reviewer for their comment. The Figure, in the light of the new data, has been replaced with new **Figure 3** (formate) and **Figure S14** (formic acid), which include analysis from simulations with a duration of 200 ns in 3 replicates. We note here that there is a slight movement of the substrate with respect to the original position, but now, it moves much less due to the stronger statistics. Any deviation is expected to be due to molecular flexibility. The contact analysis that the reviewer suggested (**Question 4.4**) strengthened the hypothesis that this density could correspond to the substrate (see **Answer 4.4**) as most stable contacts correspond to the EM density state. In addition, we never stated that this density unambiguously corresponds to the substrate, as the resolution is not sufficiently high to make this claim, but the MD data support its localization (firebrick bars in the new **Fig. 3**, **Fig. S14** and **Fig. S16** contact counts). Our revised manuscript still does not claim to resolve the substrate unambiguously.

Question 4.7:

- How is the pink residue 209 bar so far from the starting point (red line)? It looks like 6-7 Å but the panels in C look far closer, indeed that the molecule is in contact, so only 3-4 Å away. Does the His contact the molecule?

Answer 4.7:

We apologize for the previous confusing panel, which has now been replaced by **Fig. 3** for the formate and **Fig. S14** for the formic acid. As we have performed the simulations the reviewer suggested, **Fig. S16** now shows that the H209 residue is very close to the substrate, and contacts the molecule. See also **Answer 4.6**.

Question 4.8:

- the response states: 'Our MD simulations corroborate the presence of formate in the density because its residence times close to residue 209 are higher'. What residence times? I can't see anything like this in the manuscript. His-molecule distances would in part address this, but residence times are very difficult to model in MD, and typically require sampling of multiple binding and unbinding events.

Answer 4.8:

We agree with the reviewer that modeling residence times is not a trivial task, and this was an unfortunate choice of words. We now avoid stating "residence times" or "residence" within the manuscript (e.g., "Additionally, the only state where the formate residence..." was changed to "Additionally, the only state where the formate location..."). For His-substrate contacts, please see Answers above (**Answers 4.4** and **4.6**).

Question 4.9:

- I've read the results section a few times and I can't tell from the manuscript what the main conclusion is. They state that there's preferential retention of formic acid (agree) which is the

opposite of the rebuttal. But then they go on to analyse the formate displacement and not formic acid. So which state of the molecule does the MD support?

Answer 4.9:

We thank the reviewer for their comment. After this comment was made, we immediately realized the confusion. In the field of formate transport, the substrate is termed as “formic acid”, even though the prevalent molecular species in the cytoplasm, for efflux, is its anion (formate). Therefore, we simplified the MD analysis to focus on formate and efflux. We have clarified this issue in the revised paper. Firstly, we added the following sentence in the Results: “Note that under physiological conditions formic acid is present in its anionic form, formate. However, we simulated both states for the sake of completeness for efflux analysis from the simulations (**Fig. 3, Fig. S14A-D**)”. To make things clear, our simulations support formate to be the species for efflux, as it is easier to “escape” the interactions with residue 209 as compared to formic acid (**Fig. S14**). Formic acid, however, can play a role in influx, which is not the focus of these simulations. We have slightly revised the Results describing the MD to reflect these clarifications.

Overall, the main highlights of this paper are the following:

- (1) Highest cryo-EM resolution for native FocA and first mapping of the entire *E. coli* channel structure
- (2) First structure of the H209N variant, which is very important for understanding translocation; here, MD provides the hypothesis, together with functional data (**Fig. S11**), that the density observed at residue 209 may correspond to the substrate.
- (3) Identification of flexibility of the vestibules via 3D variability analysis of the cryo-EM data, showing ground states of FocA that include disordered and ordered termini.
- (4) The first model underlying the interaction between FocA and PflB, analyzed using flexible macromolecular docking and cross-linking MS data.

Question 4.10:

In S14 “the MD simulations corroborate the asymmetric nature of the pentamer”. Please explain. The periplasmic regions have a different RMSD than the cytoplasmic? Firstly, this can't easily be seen in the current graphs at all. Second, I don't see how the dynamics of the loops in the protein is of relevance to substrate transport at the center of the channel. Perhaps I'm missing something here.

Answer 4.10

The reviewer is either confused that the symmetry point of the C5 channel is the pore that translocates the substrate (unlikely), or argues that effects on the vestibules do not have an impact on what happens in the middle of the channel. Indeed, structurally, there might be no impact, but as we show with the 3D variability analysis (**Fig. 4**), these conformational changes can dramatically restrict the channel entry points. Also, as we explain in the manuscript the N-ter α -helix is involved in PflB recruitment, therefore, actually, critically accepting the substrate for translocation. Considering the confusion, we revised **Fig. S18** to improve clarity. We are now simply showing the median, Q95 and maximal per-residue RMSD of the 4 conditions (2 tautomers of HIS; charged state of HIS; and variant ASN) for each protomer. This highlights that we have certain regions within FocA that undergo conformational changes, indicated by higher max RMSD values. As these regions confine the channel vestibules (see **Fig. 4, Figure S17**), this is of relevance for the entire protein complex, as (a) the channel can be confined, (b) the access can be limited, (c) the protein can undergo changes allowing binding of interaction partners (e.g. PflB).

Question 4.11:

Aside from these concerns, I have some smaller changes that are needed:

- CHARM-GUI > CHARMM-GUI.

Answer 4.11:

We corrected this typo.

Question 4.12:

- Please spell out the lipid names. For instance, what is PYPG?

Answer 4.12:

We added all full names of all lipids. The text now reads:

The lipid composition for E. coli inner membrane during stationary phase was used²¹, containing 1-Palmitoyl-2-oleoyl-sn-glycero-3-phosphoethanolamine (POPE): 1-Palmitoyl-2-palmitoleoyl-sn-glycero-3-phosphoethanolamine (PYPE): 1-Oleoyl-2-palmitoleoyl-sn-glycero-3-phosphoethanolamine (OYPE): 1-Plasmenyl-palmitoyl-2-oleoyl-sn-glycero-3-phosphoglycerol (PYPG): 1-Palmitoyl-2-cis-9,10-methylenehexadecanoyl-sn-glycero-3-phosphoethanolamine (PMPE): 1-Pentadecanoyl-2-cis-9,10-methylenehexadecanoyl-sn-glycero-3-phosphoethanolamine (QMPE): 1-Palmitoyl-2-cis-9,10-methylenehexadecanoyl-sn-glycero-3-phosphoglycerol (PMPG) in a 6:17:5:7:32:8:3 ratio (78 total lipids) for the outer leaflet, and 6:17:5:12:20:8:8 (76 total lipids) for the inner leaflet.

Question 4.13:

- Note that this lipid composition is far too complex for a 20 ns simulation – indeed even a 1 μ s simulation would likely not be sufficient to equilibrate the membrane. This makes comparison between simulations more difficult, as the local lipid composition will not be the same. This is why model membranes are typically used (e.g. POPE/POPG)

Answer 4.13:

We agree that both the initial 20 ns and the revised 200 ns are not sufficient to fully equilibrate the membrane. This is also seen in the time-resolved RMSD plots (**Fig. S13**). The membrane was included simply to (a) create defined compartments for cytosol and periplasm and (b) stabilize the transmembrane protein. The reviewer can also appreciate the fact that we did not include any analysis regarding membrane interactions. We chose the published composition over a model membrane to allow future experiments and continue our simulations for deeper protein-membrane analysis. Considering these, we included the following in the Results section: “Although the complexity of the lipids used in our MD simulations limits the convergence of the membrane to full equilibration, the effects we report concern the equilibrated protein molecule. However, it will be important to investigate in future studies how this asymmetry is influenced by the membrane environment.”

Question 4.14:

- The protein chains were elongated using modeller. How were the new sections modelled – as helix, loop, turn etc? The model should be made available for download by the readers.

Answer 4.14:

The additions were done with the AutoModel class without enforcing any secondary structure. We added the following sentence to the manuscript:

[.] using the AutoModel function without enforcing any secondary structure.

The models are now available together with the MD input files, and the scripts for analyzing the data at Zenodo. The data is referenced in the Data Availability section.

Question 4.15:

- Was it CHARMM36 or CHARMM36m? This needs a reference.

Answer 4.15:

The CHARMM36m force field was used, and is now correctly named and referenced in the materials section. We thank the reviewer for pointing out this slight inconsistency.

Question 4.16:

- line 385 “manual mutation to formate” – what does this mean?

Answer 4.16:

We apologize for the confusion and we have now re-written this Methods section. We hope that addition of formate is now clear: “To add 200 mM formate ions into the cytosolic space, the water box was neutralized with 200 mM KCl to atomic coordinates, followed by extraction of the chloride ions between $-20 < z < -73$, and using PyMOLs align function to align free formate (PDB-ID FMT). During system generation using psfgen, the free formate (FMT) was aliased as FORA. Eventually, the simulation box containing both membrane systems and the cytosolic formate was neutralized by 200 mM KCl”

Question 4.17:

- “minimized” – using what algorithm?

Answer 4.17:

The default “minimization” function of NAMD2 was used, applying a conjugate gradient method. We updated the Methods section as follows:
[...] the system was first minimized for 10,000 steps using the default conjugate-gradient line-search minimizer of NAMD (function ‘minimize’) [...]

Question 4.18:

- “constrained” – how? With what approach and what force constant?

Answer 4.18:

Atoms were constrained using the NAMDs ‘constraints’ functionality. Constrained atoms were defined in a constraints-pdb file, and per-atom force constant was defined in the B-factor column. A constraint exponent of 2 was used, and the restrained atoms were assigned a value of $5.00 \text{ kcal} \cdot \text{mol}^{-1} \cdot \text{\AA}^{-2}$ in the B-factor column.

We added the following part to the Methods section: “A constraint exponent of 2 (‘consexp 2’) was used, and restrained atoms were assigned a value of $5.00 \text{ kcal} \cdot \text{mol}^{-1} \cdot \text{\AA}^{-2}$ in the B-factor field.”

Question 4.19:

- Was a 1 fs time step really used for both equilibration and production? Why was this used vs a more conventional 2 or 4 fs time steps.

Answer 4.19:

We thank the reviewer for this question. During revision, we used a 1 fs time step during equilibration, but during the 200 ns production phase, we now use the more conventional time step of 2 fs. We now report this in the Methods “... except the time step was set to 2 fs.”

Question 4.20:

- Fig 3A doesn’t add anything at all to the analysis – it’s simply an image of a simulation box. Same goes for Supp Movie 1

Answer 4.20:

We moved the original Fig 3A to the Supplementary, now Fig. S13. We also removed the Movie as the reviewer instructed.

Question 4.21:

- “custom Python script” this needs to be made available via GitHub, otherwise I can’t comment on the data analysis.

Answer 4.21:

During the revision process, our analysis workflow now includes the established Python package MDanalysis instead of custom Python code, which we describe in the Methods section “MD data interpretation”. Additionally, all Jupyter Notebooks are included in Zenodo, allowing the reviewer, and also any reader full accessibility to our workflows as described in the data availability section.

REVIEWERS' COMMENTS

Reviewer #4 (Remarks to the Author):

I thank the authors for their revisions and comments. It is great that the authors have added MD data to support their experimental analysis, and I think it bolsters the study. I think (hope) that the additional sampling and analyses improves the study and strengthens their conclusions.

The changes they have made now assuage my concerns.

I make an optional suggestion based on their newly included data:

- The RMSD of the membrane in S13B isn't a particularly useful metric. There are better ways of showing membrane equilibration, but this probably isn't that necessary for the current work. I suggest this is dropped with just the protein backbone RMSD plotted.

I would also like to raise a suggestion as feedback for future studies:

- Having two copies of FocA in separate membranes is an interesting approach. However, to my mind, these should ideally be further than 2 nm apart. Waters and ions are strongly impacted by membrane so there's a risk of weird biophysical effects occurring between two close membranes. Also, dual membrane systems can be problematic with solvent densities, ion concs etc. In the current study (looking at transport in the centre of a channel) I don't consider it an issue, but for future studies I suggest sticking with a single membrane if possible.

Answer:

We thank the reviewer for their positive assessment of our revisions and for acknowledging that the added MD data strengthen the study. Following their suggestion, we have removed the membrane RMSD from Supplementary Figure 13B and retained only the protein backbone RMSD. We also appreciate the reviewer's constructive feedback regarding the dual-membrane setup and acknowledge their valuable scientific insights for guiding future studies.